# Multifactorial Analysis to Determine the Applicability of Wind Power Technologies in Favorable Areas of the Colombian Territory

Andrés Rodriguez-Caviedes [1] and Isabel C. Gil-García [2,*]

1   Exploration and Development of Fields—JOA-ANH of Frontera Energy Colombia Corp (FEC), Bogotá 110111, Colombia; arodriguezc@fronteraenergy.ca
2   Faculty of Engineering, Distance University of Madrid (UDIMA), 28029 Madrid, Spain
*   Correspondence: isabelcristina.gil@udima.es

**Abstract:** Colombia has an energy matrix that is mostly hydroelectric and includes renewable energies such as wind power, which represents a minor contribution. The only operational wind farm is in the northern part of the country, where more projects will be implemented in the future in search of increasing the installed capacity and electricity generation. However, the wind potential and behavior of other areas of the national territory have been little reviewed. The most recommended method to characterize the potential in different areas of Colombia is to use real data, generating vertical extrapolations and respecting the good practices of the wind industry. The foregoing not only allows the generation of statistical and descriptive characterizations but also, together with the climatological, geographical, and technological variables (turbines), an estimate of the generable energy that can be obtained. In the described study, we applied specialized software to generate a rating matrix, from which it was possible to issue an opinion on five possible locations obtained from the theoretical development of micrositing, where 14 factors were reviewed. There is no published research of this nature for the country, so it is relevant in terms of novelty. Finally, it can be concluded that in Colombia, the wind potential should not be associated with a specific region, since there are data throughout the territory where this type of research can be replicated.

**Keywords:** wind energy; wind variability wind potential; onshore wind power plants; micrositing; descriptive and statistical analysis; relevant factors for optimal locations

## 1. Introduction

The great energy potential of wind resources has resulted in their increased contribution to the generation of electricity around the world in recent years, although this trend is much more marked in countries that are members of the G20. On the other hand, in regions such as Latin America and, more specifically, in South America, wind energy seems to be a very little explored renewable alternative, and although its exploitation is increasing, this is occurring rather slowly, or at least not rapidly enough as it seems is required to offset the effects of greenhouse gas emissions and climate change. These statements can be validated through the reports of EMBER [1] and IRENA [2], in which the geopolitical behavior and installed capacity for the year 2020 are dimensioned.

For example, in the 2019–2020 period, the EU 27+1 group increased its contributions to generation through wind energy by 9.8% (41.91 TWh), while China and the United States achieved an increase of 15.1% (61.20 TWh) and 14.1% (41.63 TWh), respectively (see Table 1). In this section, South America generally experienced an upward trend, since nations such as Argentina, Brazil, Uruguay, and Peru increased their contribution of electricity generation through wind projects. However, in the specific case of Colombia, the object of analysis that concerns us, a reduction of 88.7% (0.05 TWh) was achieved (see Table 2).

**Table 1.** Some representative countries and regions in which wind energy experienced positive percentage changes in annual generation for the period 2019–2020 [1].

| Country/Region | Annual Variation in Wind Power Generation (%) | Annual Variation in Wind Power Generated (TWh) |
|---|---|---|
| Kenya | 315.7 | 1.19 |
| Argentina | 88.5 | 4.4 |
| Norway | 68.6 | 3.8 |
| Japan | 24.5 | 2.1 |
| Morocco | 22.3 | 0.86 |
| South Korea | 19.4 | 0.5 |
| Egypt | 18.3 | 0.4 |
| China | 15.1 | 61.2 |
| Australia | 15 | 3.17 |
| México | 14.4 | 2.4 |
| Uruguay | 14.3 | 0.7 |
| United Kingdom | 14.3 | 9.3 |
| United States | 14.1 | 41.6 |
| Canada | 10.1 | 3.1 |
| EU 27 +1 | 9.8 | 41.9 |
| Perú | 9.5 | 0.2 |
| EU 27 | 9 | 32.7 |
| Germany | 6.8 | 8.6 |
| Denmark | 3.1 | 0.5 |
| Brazil | 1.2 | 0.7 |

**Table 2.** Some representative countries and regions in which wind energy experienced negative percentage changes in annual generation for the period 2019–2020 [1].

| Country/Region | Annual Variation in Wind Power Generation (%) | Annual Variation in Wind Power Generated (TWh) |
|---|---|---|
| Colombia | −88.7 | −0.05 |
| Costa Rica | −18.2 | −0.33 |
| Hungary | −13 | −0.10 |
| Honduras | −12.3 | −0.11 |
| Portugal | −10 | −1.37 |
| Austria | −9.6 | −0.73 |
| Bolivia | −8.6 | −0.01 |
| Bosnia and Herzegovina | −7.2 | −0.02 |
| Italy | −6.8 | −1.38 |
| India | −4.6 | −2.89 |
| Philippines | −3.6 | −0.04 |

Regarding the installed wind capacity, Colombia signed a 15-year bilateral contractual agreement in 2019 with the Portuguese company EDP Renováveis [3], for which the aim is to increase the installed capacity from 19.5 to 510 MW (values reported in the IRENA Renewable Capacity Statistics 2021) through the construction of two wind farms called Alpha (212 MW) and Beta (280 MW), both located in the municipalities of Uribia and Maicao in the department of La Guajira. These wind farms will not be operational until the course of the year 2022 when the construction processes are scheduled for completion and, thus, the exploitation stage can begin. Therefore, Colombia maintains only 19.5 MW of real operational installed capacity of wind energy, in a single department of the country, out of a total of thirty-two, which shows a development for the most favorable area in terms of wind variability conditions, but a significant gap in the opportunities that other areas can offer.

According to the Colombian Association of Electric Power Generators (ACOLGEN) [4], Colombia's electricity generation matrix fits into a renewable type due to the majority of the contribution coming from hydroelectric plants (11,846.2 MW (38.3%)), but dependence

on these necessarily implies submission to the behavior of climatic phenomena, mainly in the regions where hydroelectric plants operate. Furthermore, the equivalence is so disproportionate given that wind generation plants correspond to only 19.5 MW (0.1%) (see Figure 1).

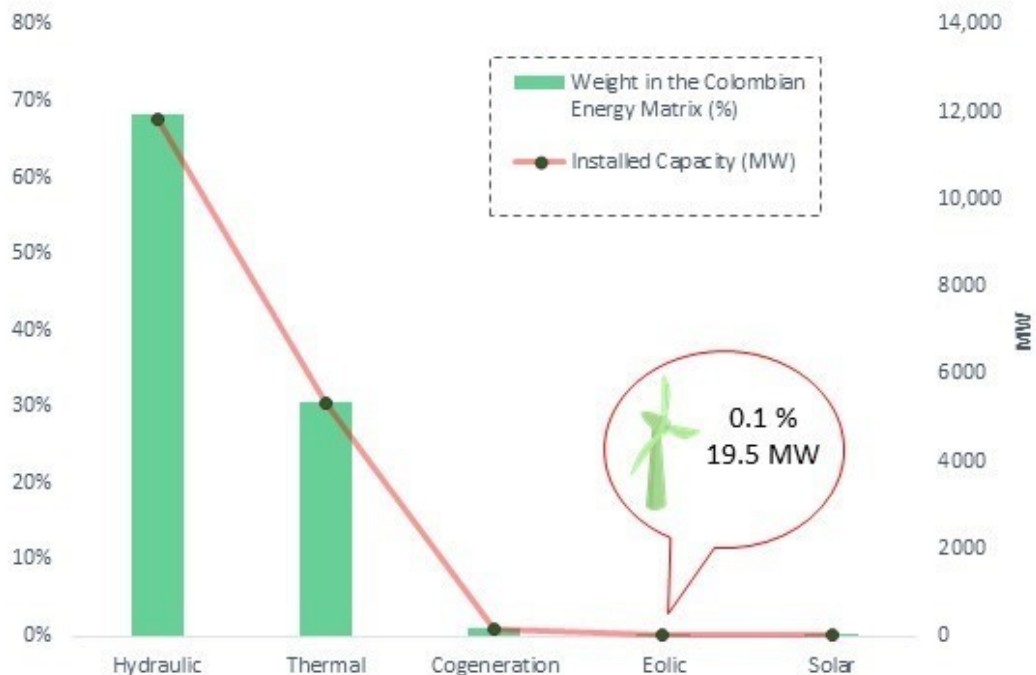

**Figure 1.** Colombian energy matrix. Own elaboration with data from [4].

South America, Central America, and the Caribbean have just 26.4 GW installed in wind projects, both onshore and offshore. This represents 3.6% of the world total, placing Latin America above the Middle East (0.1%), Africa (0.35), Eurasia (1.3%), and Oceania (1.4%) but still very far from North America (19%, including México), Europe (28.3%), and Asia (45.3%) [5].

In the specific case of Colombia, only the Jepírachi wind farm is operational and contributing electricity generation to the grid. This farm is in the municipality of Uribia in Upper Guajira, Colombia, in the so-called Cabo de la Vela sector. It is made up of fifteen 1.3 MW wind turbines developed by the Nordex company using the N60 model [6].

Jepírachi functions as a minor plant of the National Interconnected System (SIN). The rotors are 60 m in diameter and are also located 60 m above the ground. The wind farm is distributed in a double row, the first with eight wind turbines and the second with seven, with an approximate separation of 1000 m between lines. The average separation distance between wind turbines on the same line is 180 m; however, due to the conditions of the terrain, it was determined to apply variations in the separation of the wind turbines and the direction of the line, defining that it was necessary to maintain an orientation of −10° north (azimuth 350°) [6].

The location of the park (according to Figure 2) corresponds to a desert area, with temperatures ranging from 20 to 40 °C, with the latter being the maximum in the warm months (May–September) and 20.8 °C the lowest for the year in the winter period (October–January). The monthly relative humidity of this region is around 73.4% [7].

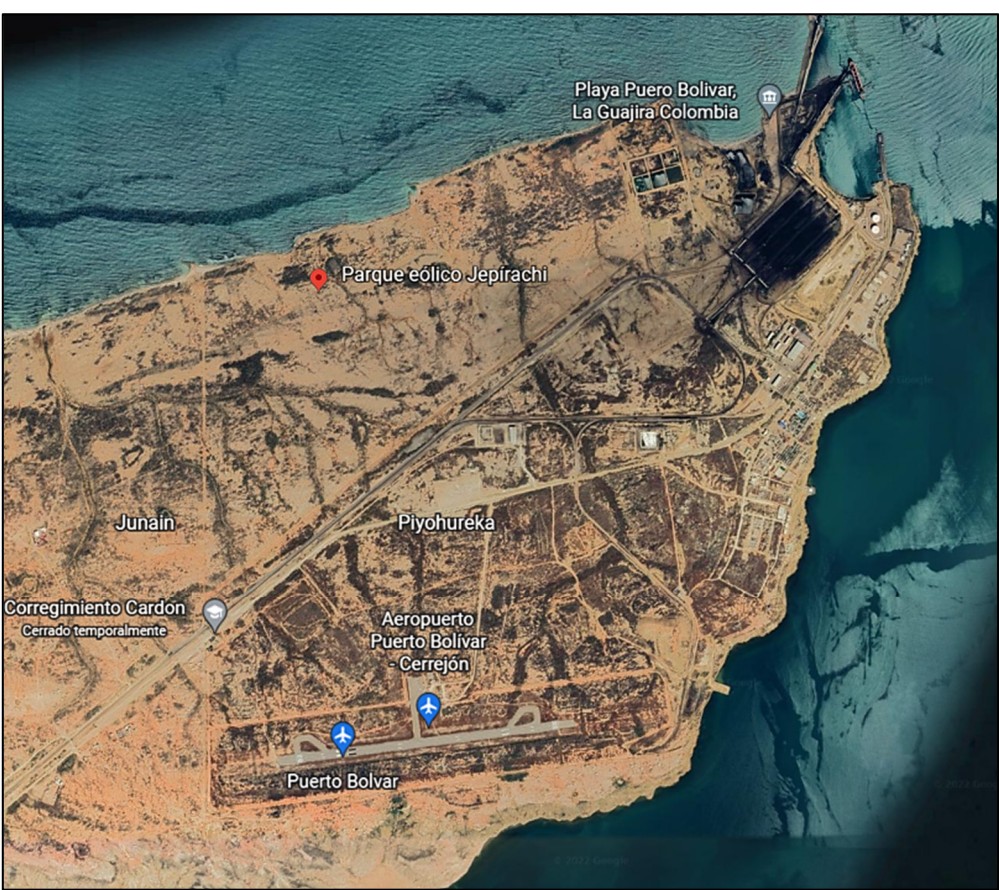

**Figure 2.** Jepírachi wind farm location. Own elaboration with data from [8].

In terms of generation, Jepírachi has experienced regular behavior. Although the capacity factor of the plant was higher than 30% in a certain year during the period 2005–2014, the equivalent average for the entire period was barely 31.9%, very low compared to the 42% projected by EPM, the owner of the project and who is responsible for the operation of the farm [9].

The current panorama of Jepírachi is not the most favorable since, according to resolution 060 of 2019 of the Energy and Gas Regulation Commission (CREG), "by which temporary modifications and additions are made to the Operating Regulations to allow the connection and operation of photovoltaic and wind solar plants in the SIN and other provisions are dictated", the wind farm is framed as not currently having the minimum technology required given its age, meaning all of its operations had to be suspended at the beginning of 2020. As it was not possible to update the equipment and facilities, EPM was able to reach an agreement with the national government on the safe operation of the park, reactivating activities in the second half of 2020 until the year 2023, when the useful life of the plant will be reached [10].

Hence, there are big questions about the suitability of renewable energy management in Colombia, specifically regarding wind energy. Why have serious design and construction projects for wind farms not been carried out in the internal regions of Colombia? Is the department of La Guajira the absolute and irrefutable owner of the wind resource in the country? Why does the diversified energy matrix have so little wind power? If there are wind data for the entire Colombian territory and even wind models for the entire country, why have multifactorial analyses not been carried out?

Although, in Colombia, there are some published studies on the characterization of wind and its applicability for new sites, there are no multifactorial analyses such as the one developed in this work. In some publications, areas were clearly reviewed with probabilistic

analyses and with alternative simulations, but none of them covered additional components to the physical characteristics such as the effects on biodiversity, society, produced energy, or economics. In addition, even if this study is limited to the most prospective areas considered by the authors, no bibliography is located where all the available data packages of the country are intended to be used to issue a matrix concept. Support for the above is extensively addressed in the Bibliographic Compilation section.

Within this ambit, the main objective of this article was to propose a viable solution to contemplate wind plant options in the rest of the country. The main contributions are as follows:

- The real available data are analyzed, the variability conditions are characterized, and alternatives of theoretical sites with energy values comparable to each other are generated.
- The analysis is carried out not only from the technical and economic points of view but through multifactorial processing, opening the door to additional discussions and complementary issues for the Colombian wind industry.

Some works related to this research but with different procedures regarding the determination of locations are listed in Table 3.

**Table 3.** Related works, methods, and countries of execution.

| Reference | Procedure/Method | Country/Region |
|---|---|---|
| [11] | Correlation Analysis<br>Meteorology Cases<br>Pearson and Spearman Correlation Maps | Colombia<br>Andean Region |
| [12] | Temporal Analysis<br>Pearson Correlation<br>Inverse Distance Weighting (IDW)<br>Static Method | Colombia<br>Boyacá |
| [13] | Reanalysis of Data (ERA5, ECMWF)<br>Polynomial Curve Fitting<br>Simple Power Prediction Model (SPPM) | Bangladesh |
| [14] | Criteria and Tolerance Maps<br>*Wasp* Software Application<br>Validation of Environmental Restrictions | Spain<br>La Rioja |
| [15] | Spatial Analytic Hierarchy Process (AHP)<br>Multicriteria Decision Making (MCDM) | Germany |
| [16] | Literature Review<br>Determination of Territorial Constraints<br>Wind Potential Based on GIS Application<br>Use of Power Curves | Spain<br>Canary Islands |
| Present work | Micrositing Technique and Multifactorial Analysis<br>*Windographer* and *Wasp* Software Application<br>Likert Scale | Colombia<br>Boyacá, Nariño,<br>Cundinamarca, Cauca,<br>and Sumapaz |

The rest of this paper is structured as follows: Section 2 describes the proposed methods; Section 3 discusses the results; and the conclusions and discussions are finally presented in Section 4.

## 2. Methods

The methodology of this work contemplates four fundamental stages, which in order of execution are bibliographic compilation, geographical and climatic analysis, definition and selection of favorable zones, and application of sensitivities.

Each of the procedures mentioned is detailed below.

## 2.1. Bibliographic Compilation

Being the first step, the bibliographic compilation is the most important, since the volume of information available about analysis is essential to delimit the scope, cover the lines of development, and build author references, which will give the investigation a realistic character and justification before each decision, and a theoretical origin.

For the specific case of our multifactorial analysis, solid databases on wind farms and projects in the country were consulted. Basically, as a result of the work described in this section, it was possible to identify that the majority of research is associated with engineering university campuses, for which details were taken from Colombian engineering thesis works, several Internet articles, and specific books on wind energy.

Initially, wind energy is energy in which the kinetic energy of the wind is transformed into electrical energy, all from structures and engineering elements designed for that purpose. It is also valid to indicate that this type of energy can be considered an indirect form of solar energy [17].

According to Cucó [18], wind is defined as a displacement of air masses caused by differences in atmospheric pressure and by the Coriolis force derived from the movement of the earth on its own axis. However, not all winds or their characteristics are usable for obtaining this renewable energy; those of greatest interest for wind farm developments are those with speeds equal to or greater than 5 m/s [12].

It cannot be overlooked that this type of work requires reliable data. For this reason, state agencies such as the IDEAM (Institute of Hydrology, Meteorology and Environmental Studies of Colombia) and IGAC (Agustin Codazzi Geographic Institute) were taken as the sources of data for climatology and geographical conditions, respectively.

On the other hand, all possible factors that interfere with the optimal selection of wind farms must be considered. In this section, we note the classification and preponderance of factors that influence a wind farm on land [19], considering as a guideline that six large groups that condition a wind project can be considered, namely, climatology, geographical conditions, socioenvironmental conditions, location, economics, and political conditions. According to the authors, these six groups have their own elements obtained through a systematic review and meta-analysis. Although these elements have been reviewed for onshore and offshore wind farms, our interest is in the first type.

The following tables, beginning with Table 4 and sequentially in Tables 5–8, show each of the six groups and the elements that were selected for the analysis according to Colombian characteristics, contemplating aspects such as data and their availability and the limitations associated with the regions.

**Table 4.** Climatology factors involved in onshore wind farms, elements, and associated reasons for research. Source: [19].

| Element | Percentage of Consideration in Taxonomic Review (%) | Reason for Selection/Rejection |
|---|---|---|
| Wind Speed | 94 | Very relevant for full characterization |
| Power Density | 12 | Needed for energy determination |
| Wind Direction | 6 | Very relevant for full characterization |
| Effective Time | 9 | Influential for turbine model selection |
| Data Availability | 3 | Vital for applying sensitivities |
| Natural Disasters | 12 | Desirable for selection or discarding of zones |
| Air Density | 15 | Necessary to geolocate real data |

**Table 5.** Geographical factors involved in onshore wind farms, elements, and associated reasons for research. Source: [19].

| Element | Percentage of Consideration in Taxonomic Review (%) | Reason for Selection/Rejection |
|---|---|---|
| Slope (*) | 71 | Needed for turbine location |
| Altitude (*) | 38 | Same as above and for air density calculation |

* Elements related to the specific geographical conditions of the reviewed polygons.

**Table 6.** Socioenvironmental factors involved in onshore wind farms, elements, and associated reasons for research. Source: [19].

| Element | Percentage of Consideration in Taxonomic Review (%) | Reason for Selection/Rejection |
|---|---|---|
| Protected Areas or Distance | 65 | Not only is it vital to consider, but it can also limit decisions |
| Agrological Capacity | 26 | Important since it is not intended to impact this industry |
| Visual Impact | 21 | Key element to avoid visual pollution |
| Noise | 24 | Relevant for risk matrix |
| Population | 12 | Relevant for risk matrix |
| Land Use | 35 | Associated with the agrological aspect |
| Flora and Fauna Impact | 35 | Relevant for risk matrix |

**Table 7.** Location factors involved in onshore wind farms, elements, and associated reasons for research. Source: [19].

| Element | Percentage of Consideration in Taxonomic Review (%) | Reason for Selection/Rejection |
|---|---|---|
| Distance/Availability of Roads | 76 | Important for zone selection |
| Distant Urban Areas | 85 | Relevant for the selection matrix |
| Distant to Point of Common Coupling (PCC) | 65 | Important to avoid isolated installations |
| Distant Transmission Lines | 50 | Important to avoid isolated installations |
| Distant Water Resources | 44 | Relevant since it includes an environmental aspect |
| Distant Industrial/Military Zones | 6 | Relevant for social and regulatory issues |

Although there are many more elements in each group, reviewable conditions were given priority on which work could be conducted. Some of the selected factors are part of the location, climatology, and sensitivity processes, while others are directly part of the selection matrix. The legal factor is not included since there is no variation in norms or regulations when changing zones in the Colombian territory.

**Table 8.** Economic factors involved in onshore wind farms, elements, and associated reasons the research. Source: [19].

| Element | Percentage of Consideration in Taxonomic Review (%) | Reason for Selection/Rejection |
|---|---|---|
| Exploitation | 29 | Necessary in any project |
| Energy Put into the Network | 26 | Necessary for selection |
| Infrastructure Cost | 24 | Needed for calculations and indicators |
| Energy Sale Price | 12 | Needed for calculations and indicators |
| Economic Contribution | 9 | |
| Payback | 6 | Required in all economic analyses |
| VPN | 3 | |
| IR | 3 | |
| Installed Capacity | 3 | Relevant and associated with total costs |

In California, more specifically in Altamont Pass, San Gorgonio, and the Tehachapi Mountains, the development of onshore wind farms can be considered to date back to the 1970s. Following this, there is evidence of installations in Europe in countries such as Denmark, Germany, and Spain. For the initial and subsequent developments, a growing trend regarding unit power began to be observed, going from hundreds of watts to kilowatts and megawatts [20].

It is also important to indicate that this work is based on the probabilistic and deterministic methods of wind speed predictions, a detail of great impact in the development of wind farms, even if the calculations are performed by specialized programs. A consolidated set of elements and studies linked to this problem can be listed: wind forecasting using least square support vector regression, state of charge at time (SoC) estimation based on an energy reservoir model, battery aging and operational cost models, and, most relevant of all, wake management for wind farms [21]. This reality is directly related to the correct use of battery energy storage systems (BESSs) and the economic impact on wind farm projects considering lost wakes as a focal point. There are at least 2 relevant works related to similar mathematical situations: lidar-assisted wake redirection [22], and adaptive control approaches [23].

*2.2. Geographical and Climatic Analysis*

First, and before starting the geographical and climatic analysis, we must bear in mind that not all wind resources are suitable for power generation. There are certain conditions of variability that must be guaranteed; therefore, in this research, a definition related to wind speed and its possibilities for energy use was used [24]. We graphically summarize this definition in Figure 3, originating from Clancy et al. (1994) and used in the work of Mendoza, where it is related to the feasibility relationship and wind speed (m/s) for energy purposes at a height of 10 m.

Colombia has two coastal regions throughout its territorial extension since it borders both the Pacific Ocean and the Atlantic Ocean (tropical Caribbean Sea). This geographical and climatological characteristic allows the country to be classified as one of the best projected in terms of the development of offshore wind potential, since its maritime extension is significant, and the diversity of its currents would allow it to have a broad portfolio of offshore projects. Despite the attractiveness of the offshore potential, only the development of wind farms on land is contemplated in this research, where the simulation process, obtaining the descriptive statistical models of the wind, and the calculation of the projected energy to generate are determined according to mathematical procedures that are widely used for high representativeness of the results

According to the Colombian foreign ministry [25], land borders were defined in a process spanning more than 100 years, in which the international limits with the neighboring republics of Brazil, Ecuador, Panamá, Peru, and Venezuela were determined.

Figure 4 shows a generalized map of Colombia, its geographical limits, and the distribution of its departments, which correspond to 32 regions. As mentioned in the introductory section, only the department of La Guajira has wind farms that are either operational or in the process of being executed.

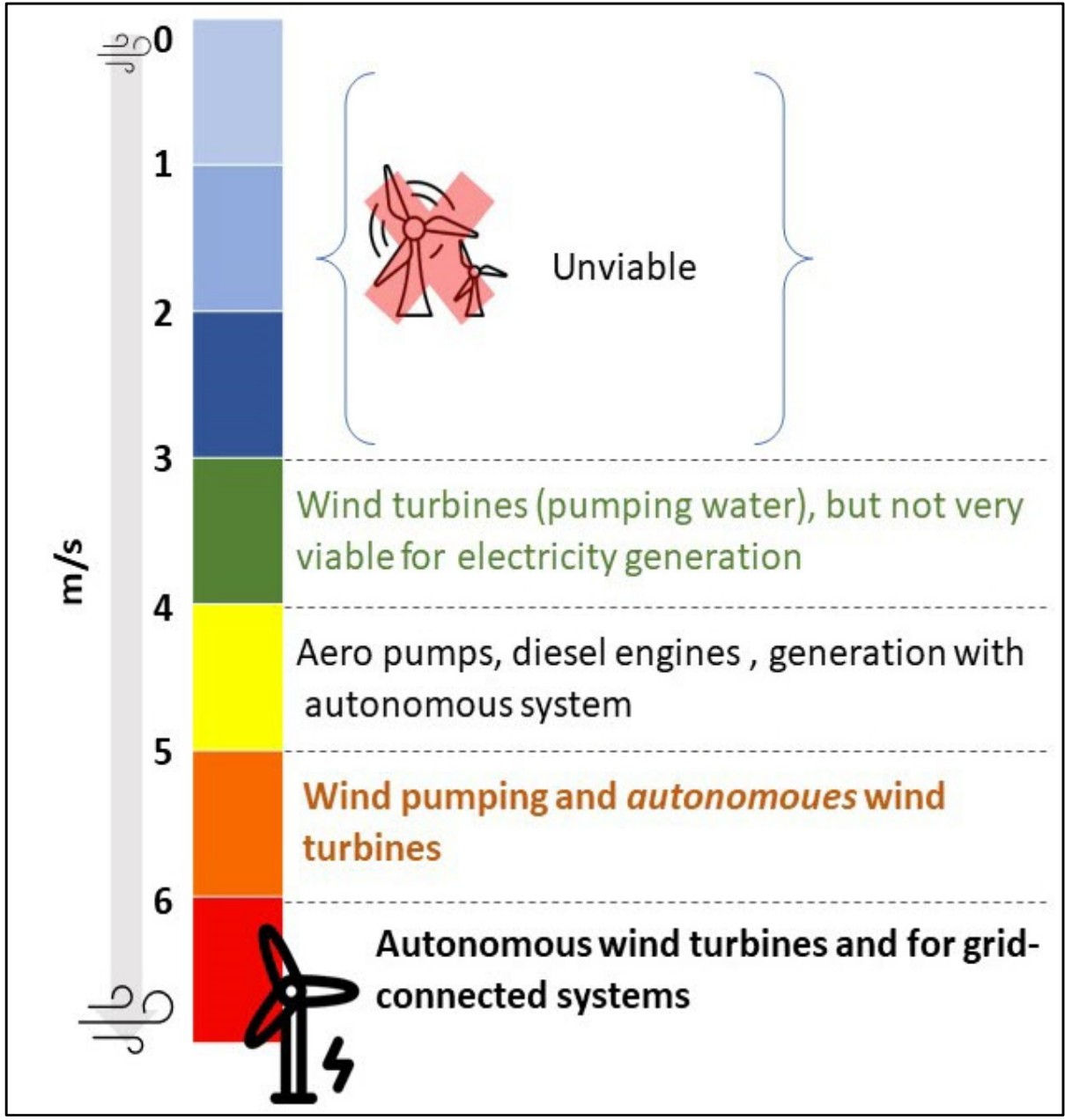

**Figure 3.** Feasibility relationship and wind speed for energy purposes. Own elaboration with data from [24].

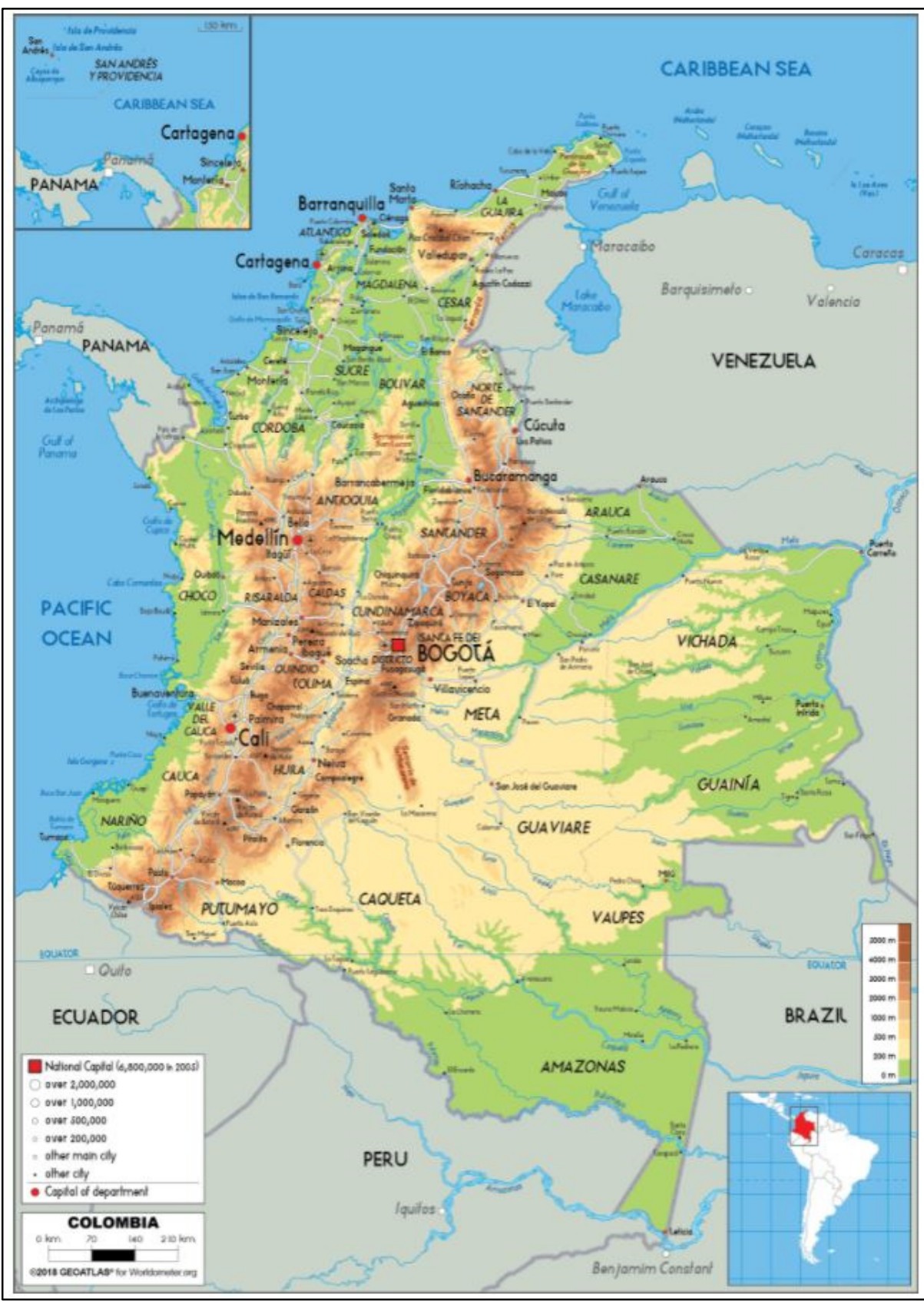

**Figure 4.** Physical map of Colombia. Own elaboration with data from [26].

In Colombia, there are no marked seasons as it is in the equatorial tropics, but there are rainy–dry seasons and climatic variability phenomena in which there is a warm phase (El Niño phenomenon) and a cold phase (La Niña phenomenon). In the former phase, apart from the increase in temperature during the day, the amount of ultraviolet radiation increases, thus increasing the hours of sunlight added to the drastic decrease in rainfall. In the latter phase, although the opposite effect is not identical, the reverse trend is very significant [27,28].

If we return to the fact of the country's low wind capacity, it is quite inexplicable how the entry of renewable energies other than hydraulics has not been massively prioritized. The Caribbean Sea and its territorial maritime limits together with the department of La Guajira have the highest wind speeds in Colombia (values close to 10 m/s), but there are also other interesting areas in the country's geography, such as the Cordillera Central (SO) in the departments of Cauca and Nariño (2500–3200 m.a.s.l.), in the center (C) and northwest (NW) in the departments of Boyacá and Cundinamarca (2400–3500 m.a.s.l.), and in the regions of the departments of Norte de Santander and Cesar (500–2400 m.a.s.l.), where speeds range between 5 and 7 m/s [11].

According to IDEAM [29], if we are to discuss precipitation, it can be affirmed that the department of Chocó, which is limited to the west by the Pacific Ocean, presents the highest annual rainfall values (5000–9000 mm), with specific wet areas with values even above 11,000 mm. On the contrary, in the department of La Guajira (where the Jepírachi wind farm is located) located in the NNE bordering Lake Maracaibo and Venezuela, annual rainfall does not exceed 500 mm. Still, in short, it can be said that the annual average rainfall is between 1000 and 3000 mm.

Regarding temperature, IGAC [30] provides climatic zoning where it is evident that the country tends to be warm to temperate in its outer limits (18–24 °C) and cold or very cold in the interior regions (6–10 °C) that are close to the Andean Mountain ranges and peaks.

### 2.3. Definition and Selection of Favorable Zones

As a first aspect, mention should be made of the geographical areas of Colombia that will not be considered in this analysis, since the characterization of all the available data on wind speeds and directions in the Colombian territory would require an extensive investigation.

To identify the excluded areas (and incidentally the areas of interest), we used the web tool of the Institute of Hydrology, Meteorology, and Environmental Studies (IDEAM) called Interactive Atlas of Winds [29]. This tool allows visualizing the geographical map of Colombia with its maritime and land limits and departmental borders, to which filters can be applied to obtain the monthly and annual behavior of variables such as the maximum and average wind speeds. In this application, it is also possible to review the estimated values of the density of wind potential for each of the 32 departments of the Colombian territory.

As a first measure within the development of the analysis, a filter was applied for the variable of the average wind speed at a height of 80 m. Although this height is lower than that where the turbines of a generator are usually located, we know that, according to theory, we will have an increase, even moderate, in the wind speed at a higher height, which justifies our choice of a height of 80 m as a minimum reference height.

The choice of this variable is clearly related to the identification of wind speeds by department, since in this way, the areas that do not initially have significant wind potential can be determined. For our multifactorial analysis, we also applied a filter where, at a height of 80 m, the department has values equal to or greater than 5 m/s in most of its territorial extension, to guarantee the possibility of having robust data coming from several possible locations (starting from the location of the measurement stations).

Figure 5 shows the initial determination of areas of interest and those excluded for analysis, following general technical considerations that allow zonal grouping throughout the territory of possible regions conducive to a wind farm. The procedure implies that within the territorial extension of these departments, a representative station can be in

terms of wind data, where some preliminary review methods can be applied, to finally frame a work polygon.

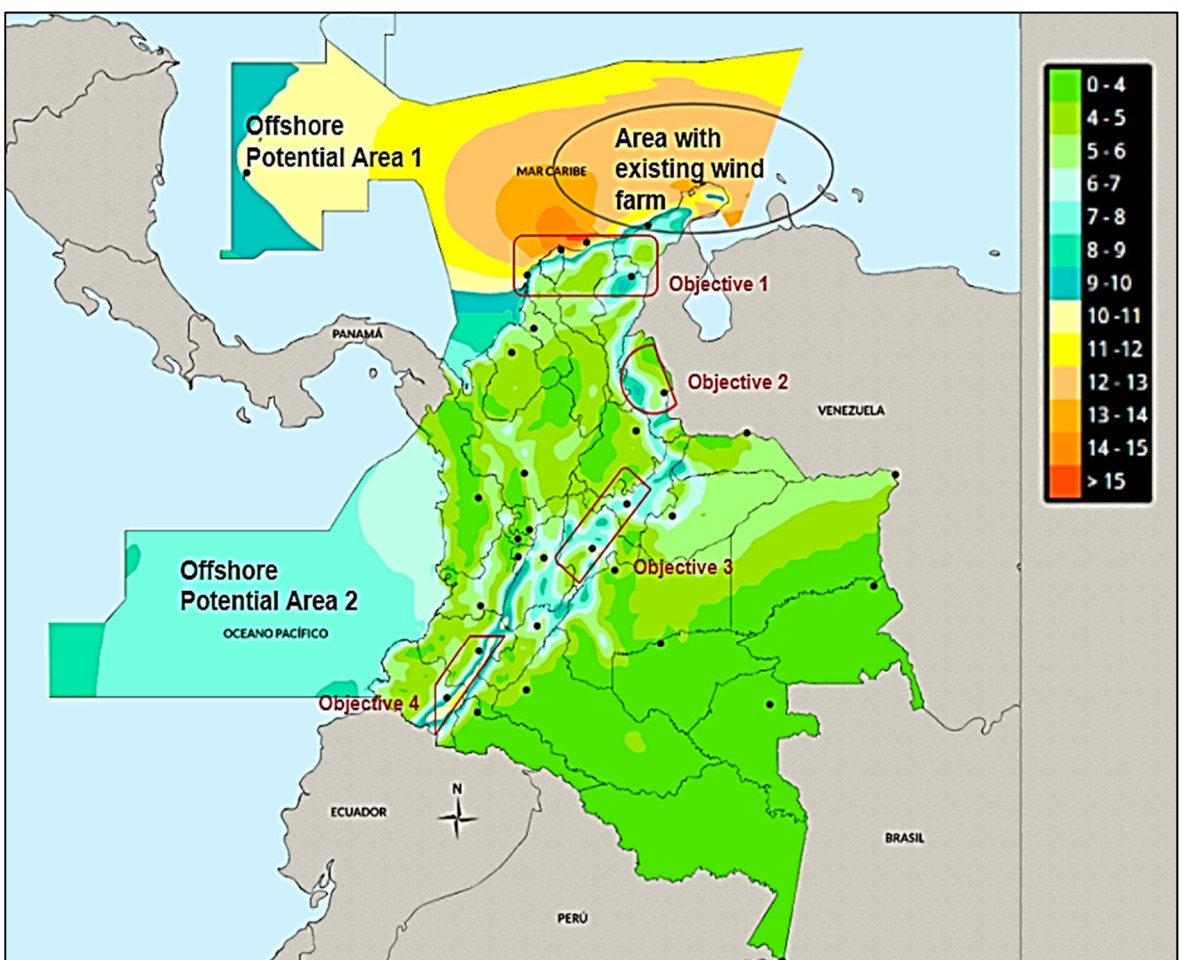

**Figure 5.** Determination of areas. Own elaboration with data from [29].

Areas where there is already exploitable wind power should be excluded, so the department of La Guajira was not considered in this analysis, since it is there where the Jepírachi wind farm is operational and where there are various projects awarded for future exploitation. In addition to this, there is excessive scientific evidence supporting the favorability of said area for wind energy exploitation [31–34], so delving further into this area would not provide greater added value.

Returning to Figure 5, the areas within the departments of interest can be observed, and those that were excluded due to geographical and technical conditions are highlighted. This initial screening indicates that at least nine departments: Boyacá, Cundinamarca, Cauca, Nariño, Magdalena, Bolivar, César, Atlántico, and Norte de Santander, and the region near Bogotá (Sumapaz) would have the indicated conditions (wind speed, location on land platform, and exempt from the presence of previous wind farms) for electricity generation.

In these areas, it was possible to determine, thanks to the information provided by the IDEAM User Service Office and the open data web of the Colombian government [35–37], that there are up to 59 active measurement stations that have wind speed and direction data at ten-minute frequencies, all recorded at a height of 10 m, this being where the climatic and agrometeorological stations are positioned. As we applied vertical extrapolation, the annual average speed limit was taken as 3 m/s since these values are at a height of 10 m and the wind turbines would be notoriously higher.

However, for accurate analysis, much more refinement and screening are needed, which were obtained from a detailed process to limit the measurement station in rectangular polygons and an area of influence around where the wind turbines could be located, all from the MAGNA SIRGAS geographic coordinates. Figure 6 outlines the processes carried out until the final five areas of interest were determined, listed in Table 9.

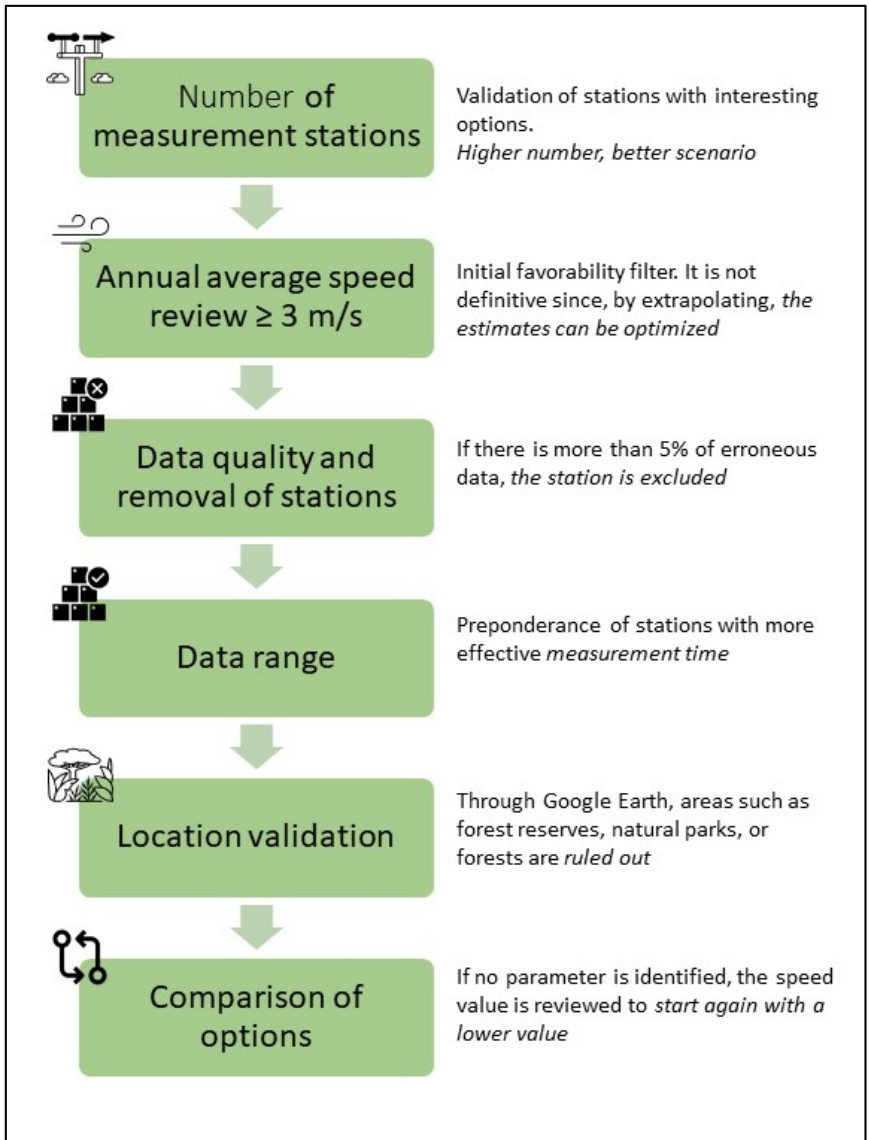

**Figure 6.** Selection scheme for areas of interest. Own elaboration.

**Table 9.** Measurement stations and departments selected for analysis.

| Department | Station Name | Annual Average Speed at 10 m (m/s) |
|---|---|---|
| Boyacá | *Toquilla* | 3.41 |
| Cundinamarca | *Santa Cruz de Siecha* | 2.5 |
| Cauca | *El Tablazo* | 1.54 |
| Nariño | *Viento Libre* | 3.64 |
| Sumapaz (near Bogotá) | *Villa Teresa* | 4.7 |

The wind speed and direction data provided by IDEAM in these selected stations do not have standard deviation values, and therefore this work does not include an analysis of turbulence. Additionally, it is necessary to emphasize that the data correspond to automatic

stations, usually with telemetric technology, which means that their reported values are preliminary and not definitive, which is important when considering the generality of the results.

The considerations used for the selection of zones can be summarized as follows:

- The polygon cannot be located outside the municipality or department of the selected measurement station.
- The area for the location of the wind turbines must correspond to the surroundings of the coordinates of the station and cannot be linked to a reserve, protected land, an indigenous settlement, or archaeologically restricted areas.
- It must also be ensured that the positioning is as far away as possible from urban areas and roads.
- The recommendations associated with good practices for a micrositing process are to position the wind turbines perpendicular to the main direction of the wind, at a distance of at least three to four rotor diameters between wind turbines in the same row and at minus five (even seven if the frequency rose is multidirectional) between rows.
- For our case, the coordinates of each station were referenced, and the polygonal points were visually located using Google Earth. This is important since their topographic files were generated in .dxf using the QGIS program.

## 2.4. Application of Sensibilities

Once the five areas of interest were determined, which initially respect environmental and social aspects in general, we moved on to the process of sensitivities and the use of specialized software. In this way, we must bear in mind that the data require refinement and detailed treatment in order not to incur significant errors. Some aspects were reviewed, and others were applied, according to specific recommendations, to the micrositing process [13,38,39].

The first procedure was to enter the organized data of each measurement station into the *Windographer©* software. This allowed us to graphically visualize the behavior of the wind at the original measurement height of 10 m. Even so, for our analysis, it was determined that our wind farm should have turbines at least 120 m high; therefore, vertical extrapolation of our data was carried out with the help of the software.

Knowing that the vertical profile of the wind varies depending on the orography and the possible obstacles present in the flow, Figure 7 supports the choice of a height greater than 100 m to avoid said variations and guarantee the fastest speeds.

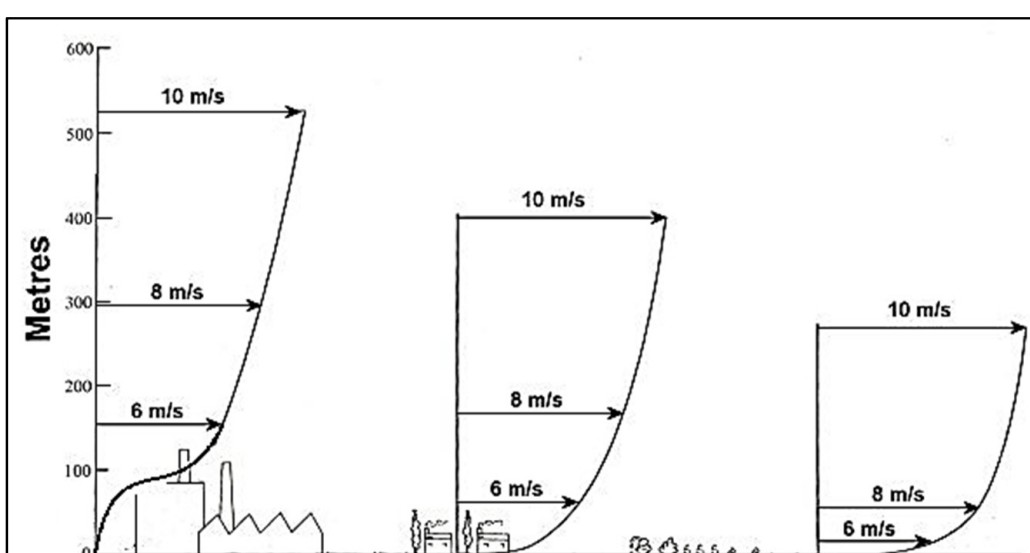

**Figure 7.** Behavior of the vertical wind profile [40].

Extrapolation gives us two determining resources for our investigation: the data at the required height of our sites for later use in the Wasp© software for energy determination, and the necessary graphs to issue a concept in statistical and descriptive terms, which we will see in detail in the Results section.

With Wasp©, it is not sufficient to use only the data extrapolated to our location height, and the topography components, geographic coordinates, and technical data of the wind turbines are also needed to be able to simulate the wind field and generate an energy estimate. Figure 8 sequentially exposes the files and steps necessary for this sensitivity and therefore for obtaining the results for each of the five selected areas.

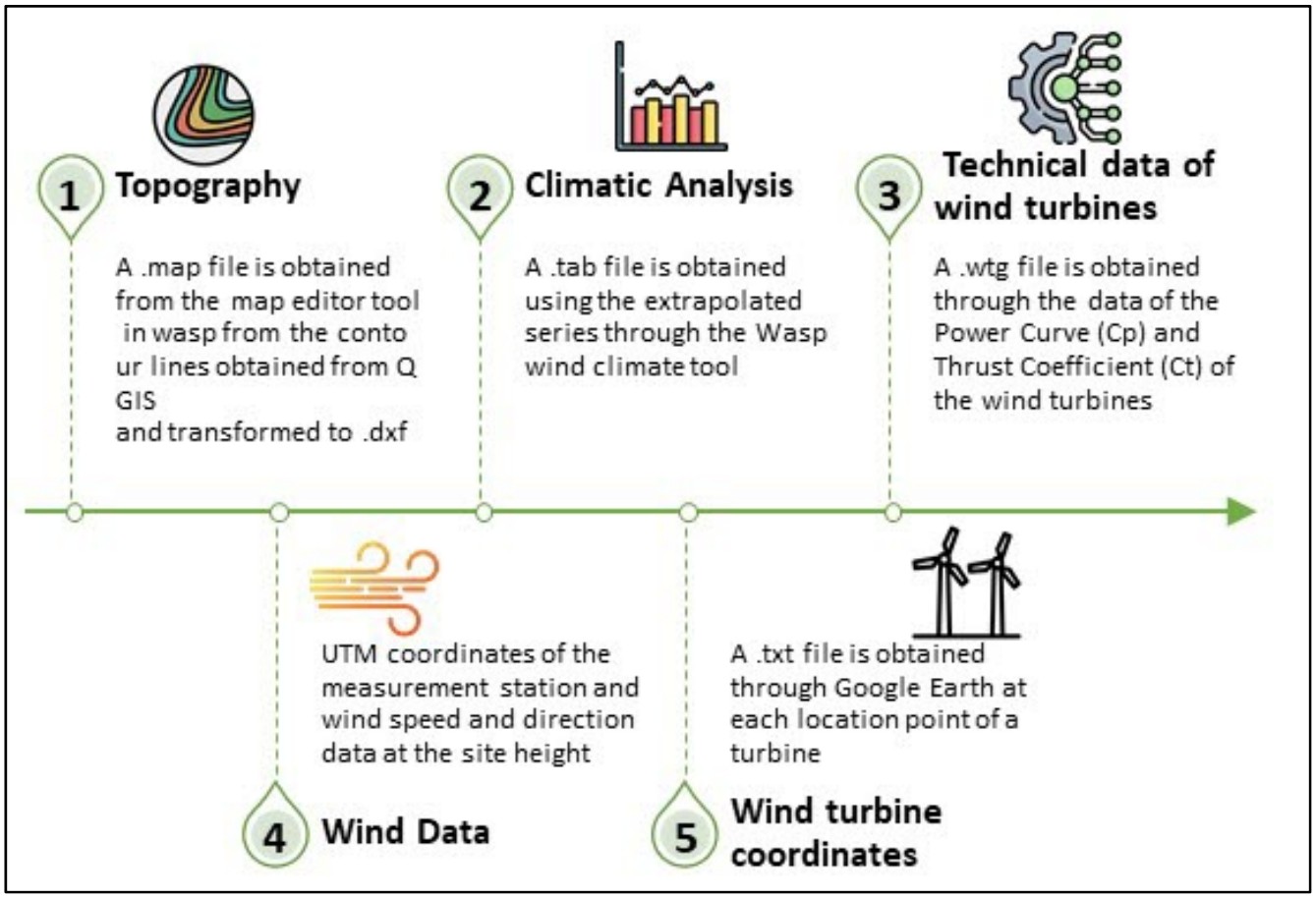

**Figure 8.** Sequence for sensitivities in Wasp. Own elaboration.

Another aspect required for the application of energy simulations corresponds to the definition of the wind turbine model. For our analysis, the *Windographer* turbine catalog was used, limiting the options to those models with nominal powers between 3 and 5 MW, which were new or modified versions from years after 2013 and from manufacturers recognized at an international level. Due to the availability of information, the V136 model (3.45 MW) of the manufacturer Vestas was selected, from which the data for the power curve (Cp) and thrust (Ct) could be obtained.

It is important to indicate that the nominal power of all the sites is also limited to 40 MW to double the installed capacity in the Jepírachi wind farm, and it is necessary to locate a minimum of 2 wind turbines and a maximum of 11, depending on the polygon built, its area, orography, environmental limits, etc.

Finally, Table 10 reports other technical aspects necessary to cover all the premises prior to the process of simulation and positioning of turbines.

**Table 10.** Parameters and conditions stipulated for sites.

| Parameter | Condition | Value (m) |
|---|---|---|
| Distance between rows | 5 times the rotor diameter | 544 |
| Distance between wind turbines | 4 times the rotor diameter | 680 |
| Site altitude | Highest position of the quadrant respecting the perpendicular | Depends on the location |
| Distance to roads | The largest possible that fits the other conditions | 1.000 |
| Distance to urban zones | The largest possible that fits in the polygon | 1.500 |
| Environmental [41–46] | Excludes reserve areas, productive agricultural land, bodies of water, and indigenous reservations | Depends on the polygon |

*2.5. Mathematical Definitions*

Remembering that the main objective is to obtain an energy value per site, most of the calculations were either associated with the extrapolation in *Windographer* or the sensitivities in *Wasp*, and there are some details that require a manual procedure for determination.

The first of these is the determination of the total height of our sites, which can be expressed empirically using Equation (1) as follows [47]:

$$T_H = A_a + WTH_H \tag{1}$$

In Equation (1), $T_H$ represents the total height of the site, $A_a$ is the average altitude taken from our level curves for each site, and $WTH_H$ corresponds to the wind turbine hub height, all of them in meters (m). In our case, this last value always corresponds to 120 m, and we only varied our altitude according to the geographical position that we analyzed.

This is a key piece of information since, in *Wasp*, this value must be entered together with the average temperature of our location for the automatic determination of the air density, simulated with the wind density tool.

Another important consideration concerns wind turbines, since although we have the power coefficient (*Cp*) and thrust coefficient (*Ct*) tables, these values are standard for predetermined air densities; therefore, to satisfy the need for real values, we required a linear interpolation or extrapolation (as the case may be). For this, we used Equations (2) and (3), as follows:

$$Y = \frac{y_1 - y_0}{x_1 - x_0} * (x - x_0) + y_0 \tag{2}$$

$$X = \frac{X_1 - X_0}{Y_1 - Y_0} * (Y - Y_0) + X_0 \tag{3}$$

where $Y$ and $X$ correspond to the required value, $y_0$ and $x_0$ are the lower values, and $y_1$ and $x_1$ are the higher values [48].

As a third element, our energy simulation in *Wasp* contemplates some losses linked to the variability of the wind and the algorithms of the program tied to the location, climatology, and mechanical structure itself. Despite this, our analysis seeks to enhance and link the results to more realistic values; therefore, an additional scenario including electrical and operational losses was applied. Regarding losses, the scenario is given by Equation (4) [49]. For net production, the time at rated power and the capacity factor are given by Equations (5)–(7), related to the study of Gil Garcia [50], as follows:

$$Nf_{TL} = (1 - \%n_1) * (1 - \%n_2) * (1 - \%n_3) \tag{4}$$

$$RNp = Npw * Nf_{TL} \tag{5}$$

$$T_{RP} = \frac{RNp}{SRp} \tag{6}$$

$$CF = \frac{T_{RP}}{Th_y} \tag{7}$$

In Equation (4), $Nf_{TL}$ as a percentage (%) is the net factor with total losses and depends on the factors $n_1$ to $n_3$, these being the percentage of electrical losses, the estimated percentage of unavailability of turbines, and the percentage of guarantee of the $Cp$ and $Ct$ curves, respectively.

In Equations (5)–(7), $RN_p$ corresponds to the real net production in megawatt hours (MWh) and is directly proportional to the factor between the net factor with total losses $Nf_{TL}$ (%) and $N_{PW}$ equal to the value of net production obtained from *Wasp*, in megawatt hours (MWh). $T_{RP}$ represents the time at rated power in hours (h) and is equivalent to the relation between $RN_p$ in megawatt hours (MWh) and $SR_P$, the site rated power in megawatts (MW). Finally, $CF$ is the capacity factor of the site and is equal to the ratio between $T_{RP}$ in hours (h) and the number of total hours in a year Thy, also in hours (h).

Finally, the section corresponding to the economic analysis requires the application of the calculation of the project feasibility validation elements [51], such as the net present value (NPV), the internal rate of return (IRR), and the playback time (PBT). These are listed in Equations (8)–(10) as follows [52]:

$$VAN = -C + \sum_{i=1}^{I=n} \frac{B_i}{(1+k)^i} \tag{8}$$

$$0 = -C + \sum_{i=1}^{I=n} \frac{B_i}{(1+IRR)^i} \tag{9}$$

$$PBT = \frac{Log(1 - k * C * B)}{-\log(1+k)} \tag{10}$$

where $C$ corresponds to the cost in euros (EUR); $B$ and $B_i$ are the total and initial income, respectively; $k$ is the update rate in terms of one (%); $i$ is the specific calculation period; and $n$ is the number of years of investment study or useful life of the installation.

As this research requires choosing a parameter based on the options obtained through the software, the best solution is to opt for a selection matrix, which can be built based on a Likert scale, which is used in social sciences and market studies and comprises a series of affirmative profiles, on which "the judgment of a subject is required. The elements represent the property that the researcher is interested in measuring and may have a numerical value depending on the agreement or disagreement that is held or perceived" [53,54]. Basically, this scale is of the ordinal type, so it defines the values for the favorability or disadvantage of an element [55].

Additionally, it is very common to link the wind site selection process to multicriteria decision analysis (MCDM). A very interesting work related to this type of procedure is that of Dhiman, Muresan, and Unguresan (2019) which shows how to select the best choices out of a given lot through mathematical models, being able to choose AHP, FLDM TOPSIS, or COPRAS [56]. In the Discussion section, the applicability and decisions made in this work about this type of methodology are reviewed.

## 3. Results

The results are grouped into five sections, starting with the descriptive statistical concepts, and ending with the selection matrix.

### 3.1. Descriptive Statistical Analysis

Using the vertical extrapolation for 120 m in *Windographer*, many graphs were obtained on which analyses were carried out. In this section, plots for the *Viento Libre* site



(Taminango, Nariño) are presented and reviewed, and the statistical concepts for each site are summarized at the end of this section.

It is also necessary to indicate that not all the graphs obtained with the extrapolation in the software were used in the analysis, and only the roses of frequency, velocity, and energy and the Weibull distributions were taken.

*Wind frequency rose:*

In this rose, the predominant quadrants of incidence of the wind on the measuring devices are determined; in other words, where does the wind blow and how often? Unidirectional or seasonal behaviors can occur, so it is an important graph for the positioning of turbines. Figure 9 shows the frequency behavior obtained at 120 m for the wind resource linked to the *Viento Libre* station.

*Wind speed rose:*

This graph characterizes the average value of all speeds for each sector. It is important to emphasize that the sector with the highest frequency will not necessarily correspond to the one with the highest velocity. Figure 10 shows the velocity behavior obtained at 120 m for the wind resource linked to the *Viento Libre* station, where the velocity and frequency are nearly coincident for the same sector.

*Energy rose:*

The energy rose is a combination of the average speed and the time at which it occurs, and it is expressed as a percentage and always associated with the directional sectors. Figure 11 shows the frequency behavior, and Figure 12 shows the comparison between the frequency and the energy on the same plane for the *Viento Libre* station at 120 m.

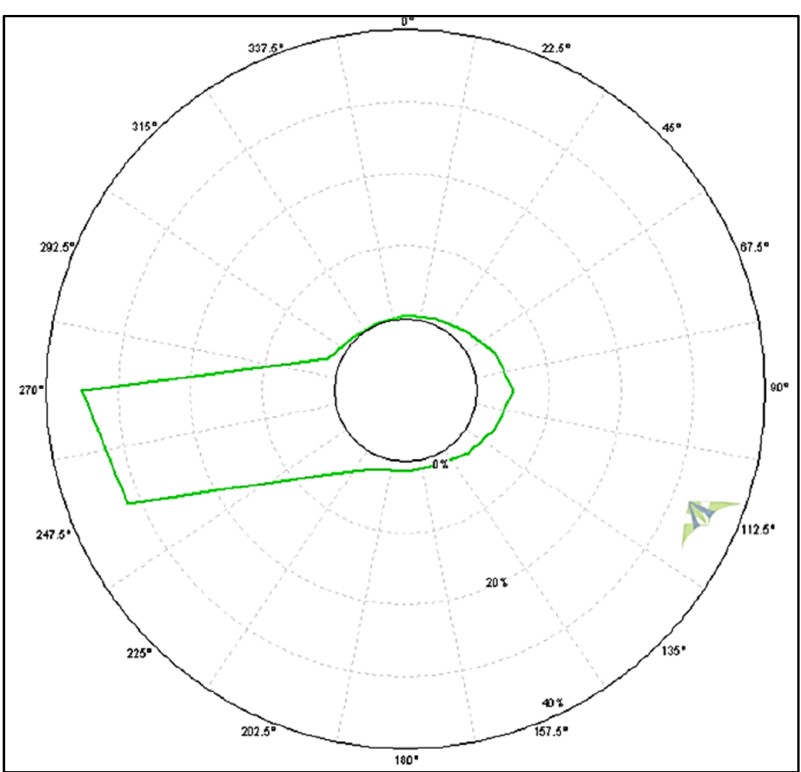

**Figure 9.** Wind frequency rose for the *Viento Libre* station. Own elaboration.

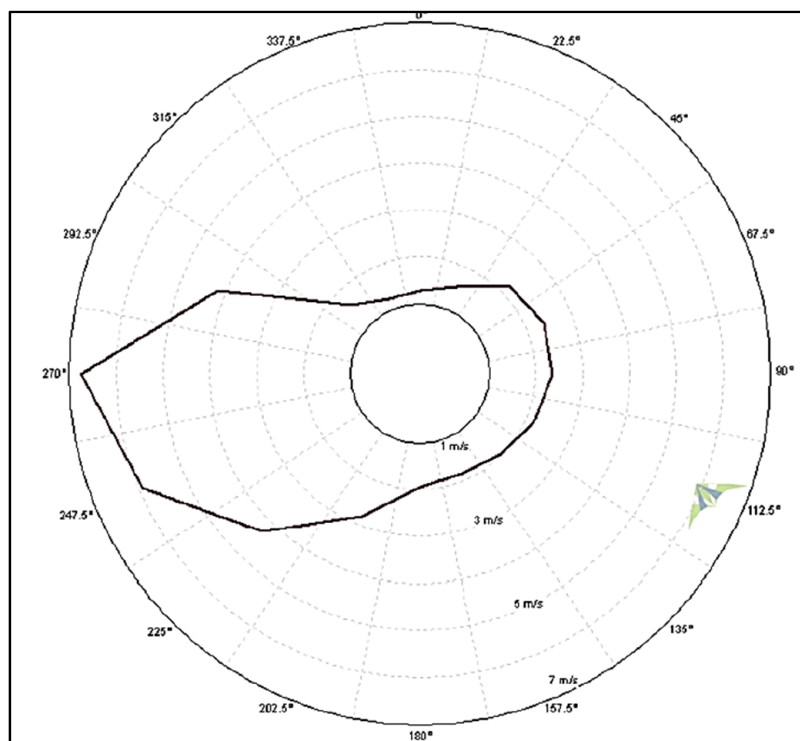

**Figure 10.** Wind speed rose for the *Viento Libre* station. Own elaboration.

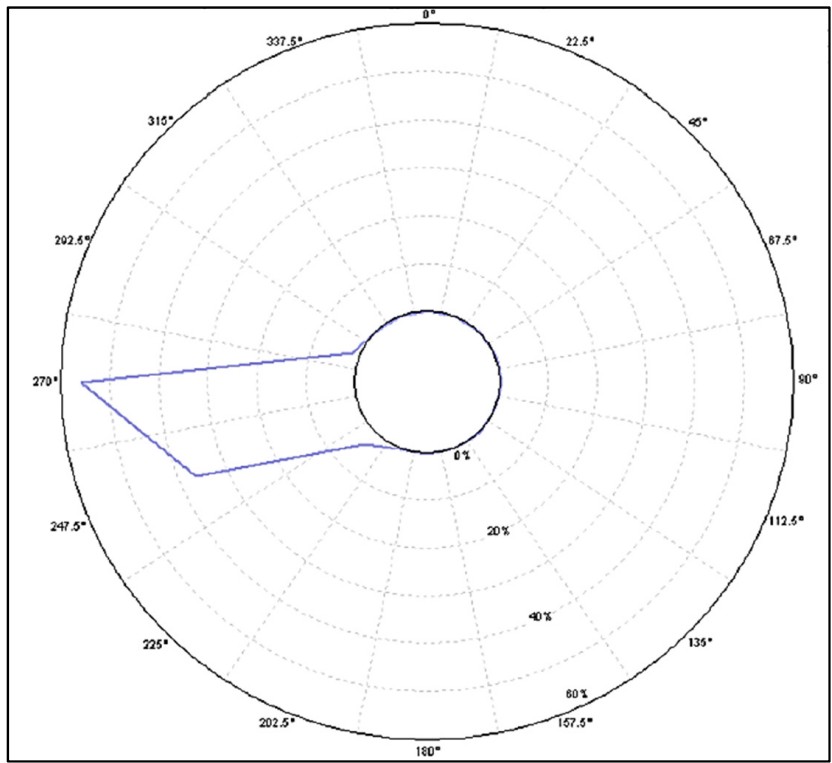

**Figure 11.** Energy rose for the *Viento Libre* station. Own elaboration.

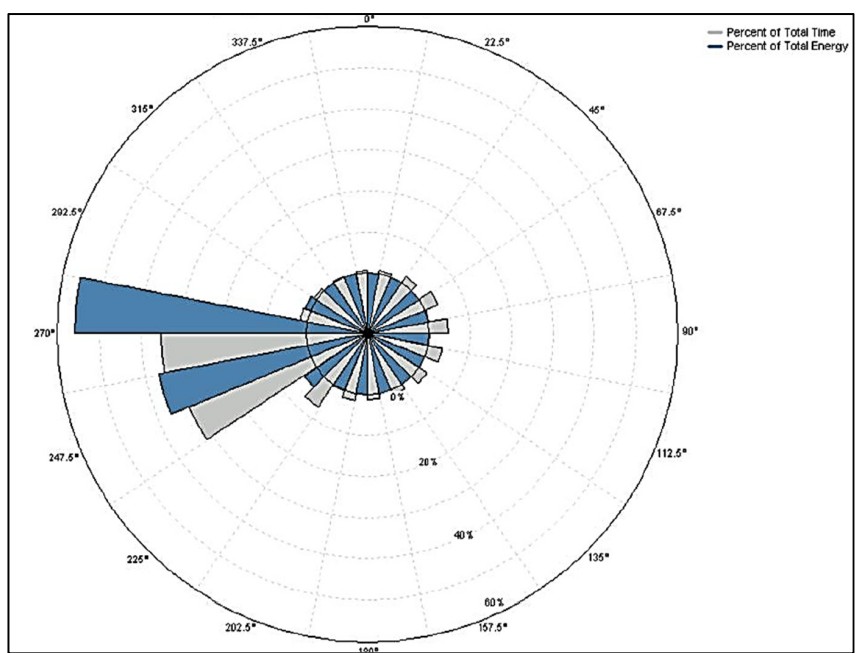

**Figure 12.** Energy vs. frequency for the *Viento Libre* station. Own elaboration.

*Weibull Distribution:*

The Weibull distribution corresponds to a statistical distribution of the data, more specifically of the linked frequency, and is considered the model that best fits the wind speeds. From said distribution, it is possible to obtain a form factor K and an average speed value A, which were *K* = 2.22 and A = 5.83 m/s for the *Viento Libre* station. The distribution is graphically depicted in Figure 13.

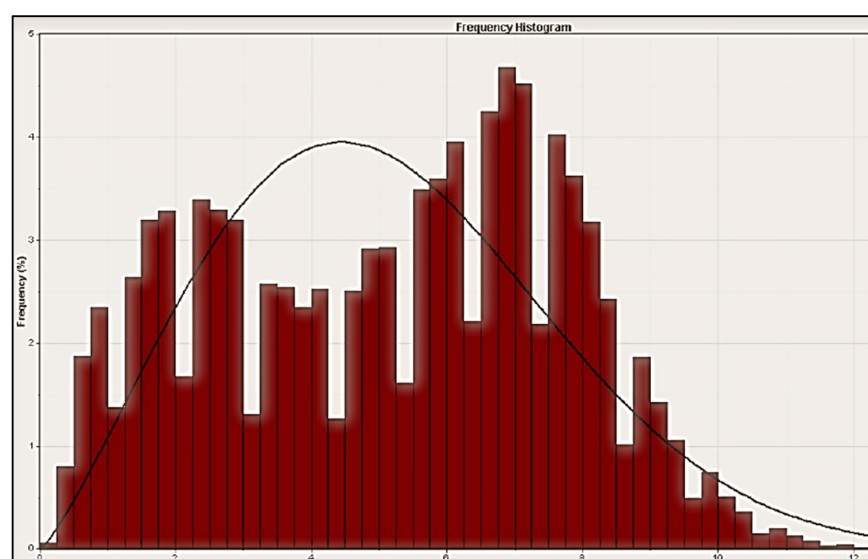

**Figure 13.** Weibull distribution for the *Viento Libre* station. Own elaboration.

With this information, together with the hourly and monthly behavior of the wind speed, the statistical and descriptive concepts of the wind resource of each site are presented below.

- *Toquilla* (Aquitania, Boyacá): Coordinates, 5°31′25″ N 72°47′27,499″ W; altitude, 2.950 m.a.s.l. The time and data range corresponds to 19 August 2017 to 1 December 2021 (4.4 years).

The main direction of the wind corresponded to ESE, with frequencies between 20% and 25% and the tendency to be directional, with very low contributions identified from other sectors. As for the wind roses, in the speed roses 100 and 120 m high, the values were close to 6 m/s predominantly in the E direction, with interesting peaks close to 12 m/s in the months of June and August. Regarding the energy rose, the energy use values were between 20% and 40% in the ESE direction.

Regarding the speed, if its behavior and daily variations in the extrapolated series are analyzed, the mornings were not very windy, and the beginning of the speed peaks was between 7 and 8 a.m. The maximum points were observed close to noon, and the daily decrease in the wind speed began around 2 p.m., so it can be said that, graphically, the behavior resembles a Gaussian bell.

- *Santa Cruz de Siecha* (Guasca, Cundinamarca): Coordinates, 4°47′3.4″ N 73°52′14,9″ W; altitude, 3.110 m.a.s.l. The time and data range corresponds to 31 December 2015 to 11 March 2019 (3.3 years).

The wind was mainly in the NW direction, with frequencies close to 24%, and it was slightly directional, with very low contributions identified from other sectors. As for the wind roses, the speed roses of the extrapolated series presented values tending to 6 m/s in a predominant NW direction, with some slight tendencies to turn toward the N or the W. The arrangement of speeds almost overlapped in all months of the year, a very valuable feature in terms of mitigating variability. Regarding the energy rose, the energy use values were between 20% and 50% in the NW direction, with the most favorable months being October, November, and December.

Regarding the behavior of the speed, if the behavior and daily variations in the series extrapolated on windy early mornings are analyzed, the start of the speed peaks occurred around 9 a.m. The maximum points were observed between 1 and 3 p.m., and the daily decrease in the wind speed began around 4 p.m. Now, if February and December are analyzed monthly, it can be found that they had higher speeds than the others, where the values decreased between March, April, September, and November, but these were not drastic falls and were compensated by growth in the month that immediately followed.

- *El Tablazo* (Popayán, Cauca): Coordinates, 2°28′29,399″ N 76°34′52,656″ W; altitude, 1.700 m.a.s.l. The time and data range corresponds to 31 December 2015 to 13 August 2021 (5.8 years).

There were two predominant directions, namely, ENE and WNW, with the first being the one with the highest net frequency (close to 14%). In both zones, the trends in behavior were very similar, only being less directional in the O direction.

As for the wind roses, in the speed roses of the series extrapolated only in the months of June, July, August, and September, the values eventually reached 3 m/s or slightly above, but the speeds were much slower the rest of the year. As for the energy rose, the energy use values were just over 20% and tended to the ESE direction; therefore, the use would not be directly associated with the frequency of direction and occurrence. It is also true that there was excessive dispersion in the monthly behavior of each extrapolated series, so the concept of wind potential for this season cannot be generic.

Regarding the behavior of the speed, in addition to being low, if the behavior and the daily variations in the extrapolated series are analyzed, the early mornings and nights were practically without wind, and there was even a decreasing fluctuation at dawn. The start of the speed peaks occurred around 9 a.m. The maximum points were observed between 2 and 3 p.m., and the daily decrease in the wind speed began around 4 p.m. Now, if it is analyzed monthly, there was only one representative peak in August, while the wind speed in the other months was very low, and the only thing that varied was its directions of occurrence.

- *Villa Teresa* (Sumapaz, near Bogotá): Coordinates, 4°20′60″ N 74°9′0″ W; altitude, 3.624 m.a.s.l. The time and data range corresponds to 19 May 2014 to 1 April 2020 (5.9 years).

The predominant direction was E (greater than 30%), and the behavior can be described as unidirectional, with almost no contribution from other sectors. As for the wind roses, in the speed roses of the extrapolated series, the possibilities of finding a high speed occurred during almost the entire year, tending to exceed 6 m/s and bordering on 12 m/s, with a semi-oval behavior. Regarding the energy rose, the energy use values were even above 60% and tended to the E direction, slightly oriented toward the ENE.

Regarding the behavior of the speed, its conditions were very interesting and conducive to turbulent conditions, and in the daily variations in the extrapolated series, there was a tendency toward inverted behavior, where the high values correspond to noon. The start of the speed peaks occurred around 2 a.m. The maximum points were observed close to noon, and the daily decrease was experienced from 2 p.m. Now, if it is analyzed monthly, it can be said that fast wind conditions were well represented between March and September with little decrease.

- *Viento Libre* (Taminango, Nariño): Coordinates, 1°37′12″ N 77°20′24″ W; altitude, 1.400 m.a.s.l. The time and data range corresponds to 23 November 2016 to 19 May 2020 (3.6 years).

The predominant direction was WSW (close to 20%), and the behavior can be described as unidirectional, with almost no contribution from other sectors. As for the wind roses, in the speed roses of the extrapolated series, the possibilities of finding a high speed occurred during almost the whole year, tending to 5 and 8 m/s. Regarding the energy rose, the energy use values were close to 60% and tended to the WSW direction.

Regarding the behavior of the speed, it does not correspond to high values, and if the behavior and the daily variations in the extrapolated series are analyzed, there was a tendency toward inverted behavior, where the high values correspond to the early mornings and nights and the valley corresponds to the mornings. The start of the speed peaks occurred around 2 p.m. The maximum points were observed after 6 p.m., and the daily decrease was experienced from early sunset to the early morning. Now, if it is analyzed monthly, it can be said that fast wind conditions were well represented between May and September.

The Weibull distributions and wind speed time-series plots are graphically presented for each site as follows (Figures 14–22). It should be noted that the Weibull distribution of the *Viento Libre* station is already presented in Figure 13.

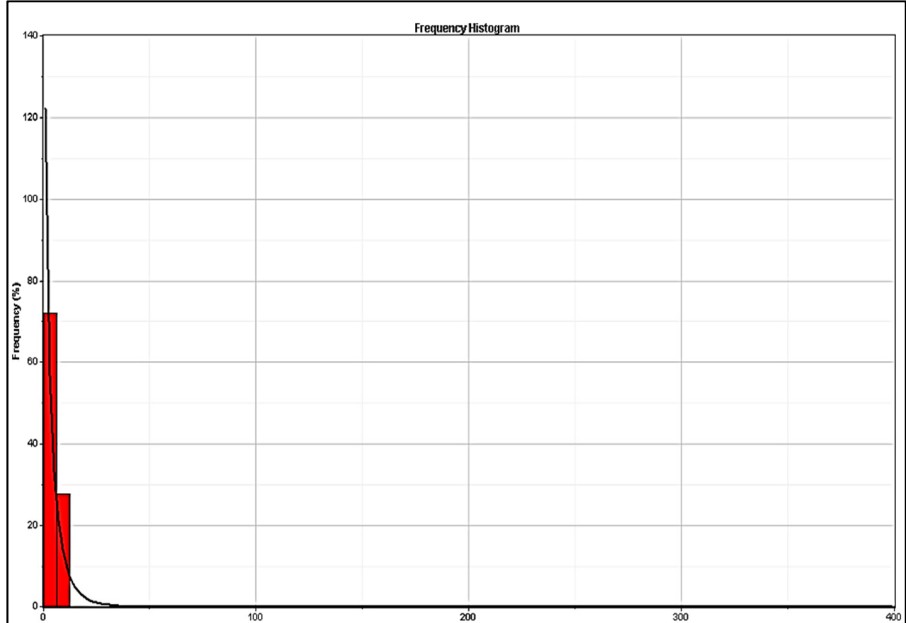

**Figure 14.** Weibull distribution for the *Toquilla* station. Own elaboration.

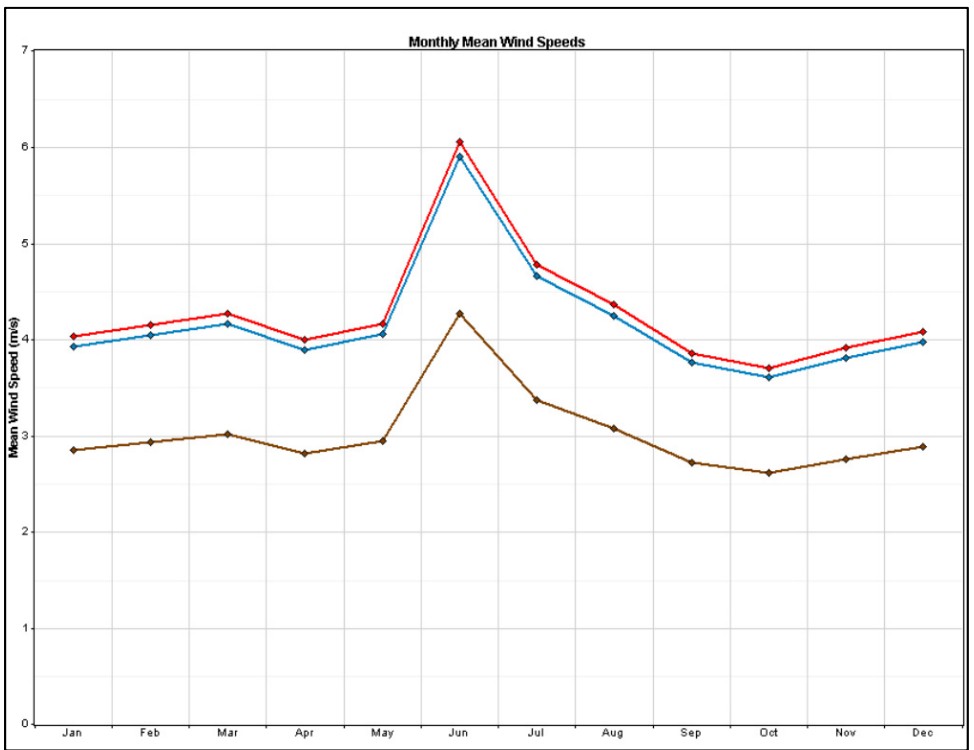

**Figure 15.** Monthly mean wind speed for the *Toquilla* station. Red: 120 m; blue: 100 m; brown: 10 m. Own elaboration.

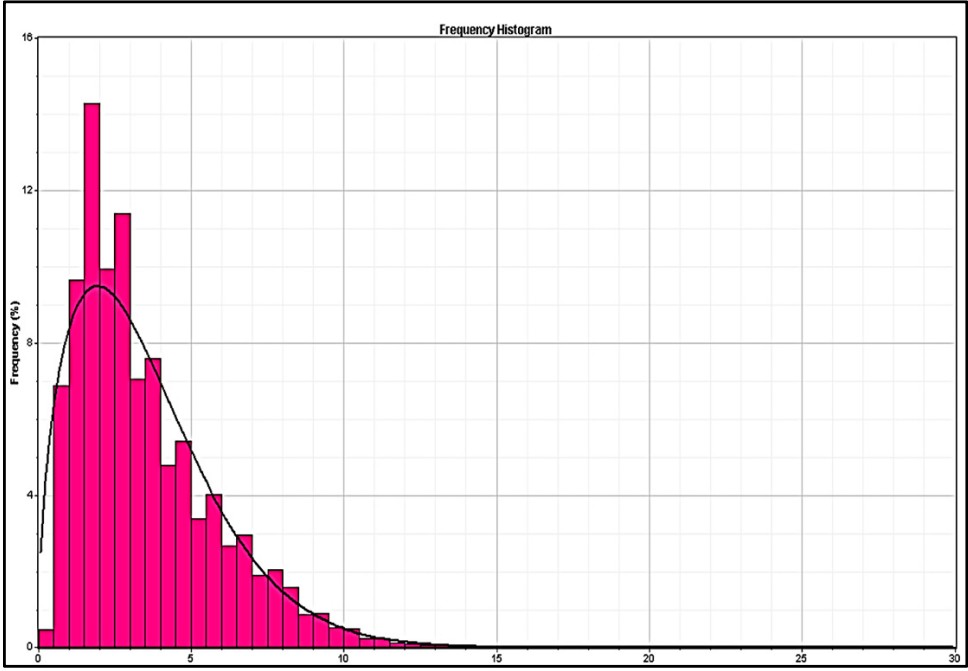

**Figure 16.** Weibull distribution for the *Santa Cruz* station. Own elaboration.

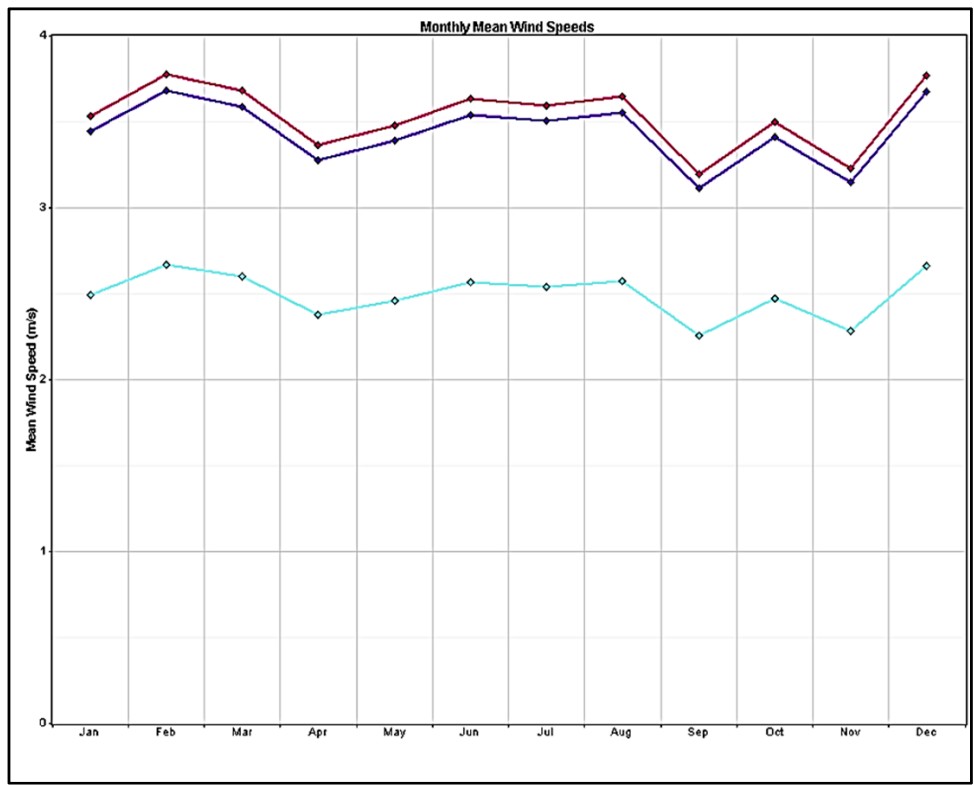

**Figure 17.** Monthly mean wind speed for the *Santa Cruz* station. Red: 120 m; blue: 100 m; aquamarine: 10 m. Own elaboration.

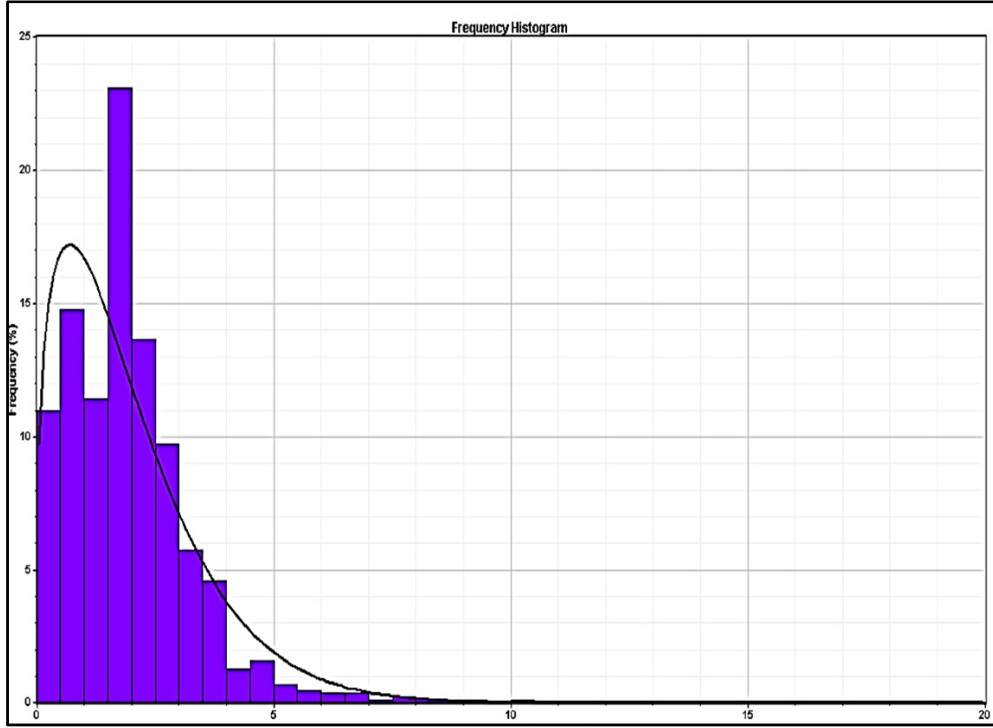

**Figure 18.** Weibull distribution for the *El Tablazo* station. Own elaboration.

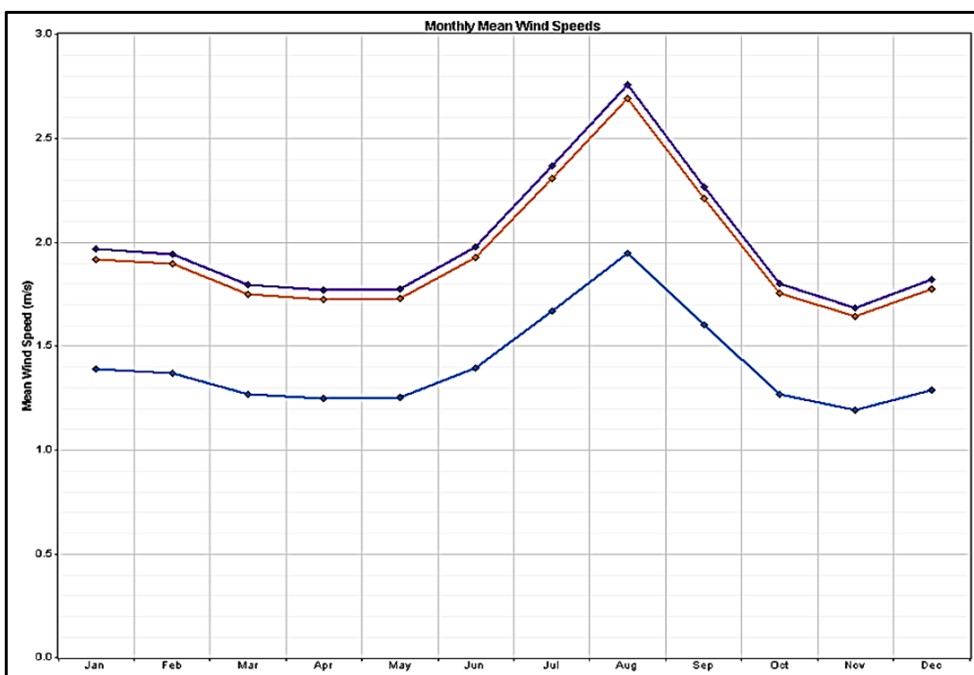

**Figure 19.** Monthly mean wind speed for the *El Tablazo* station. Purple: 120 m; orange: 100 m; blue: 10 m. Own elaboration.

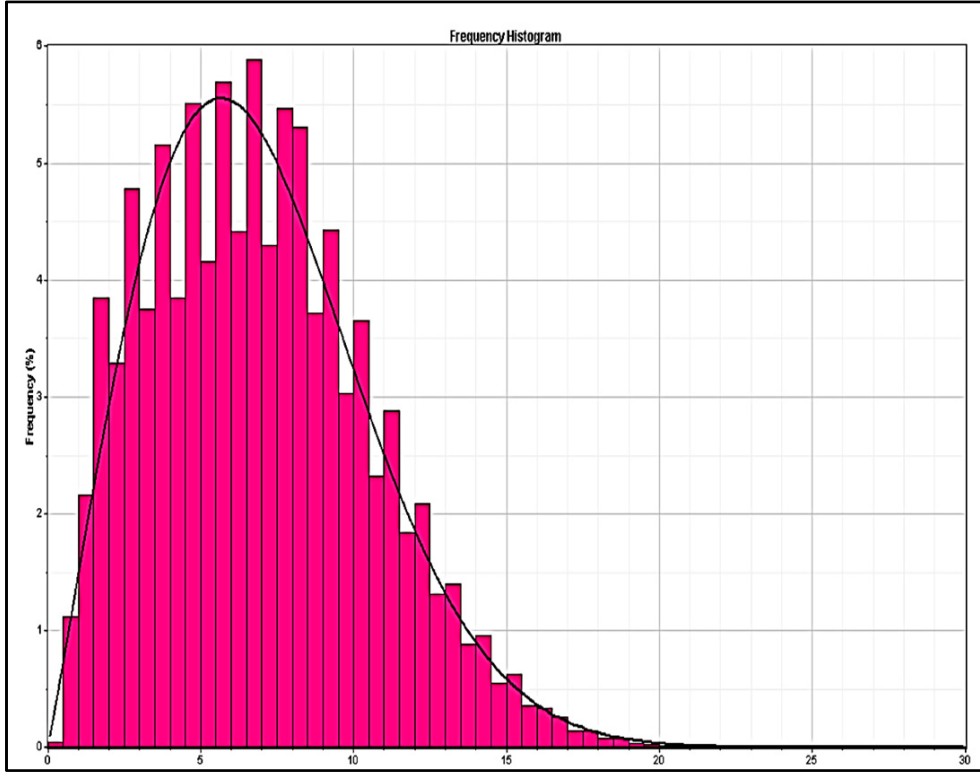

**Figure 20.** Weibull distribution for the *Villa Teresa* station. Own elaboration.

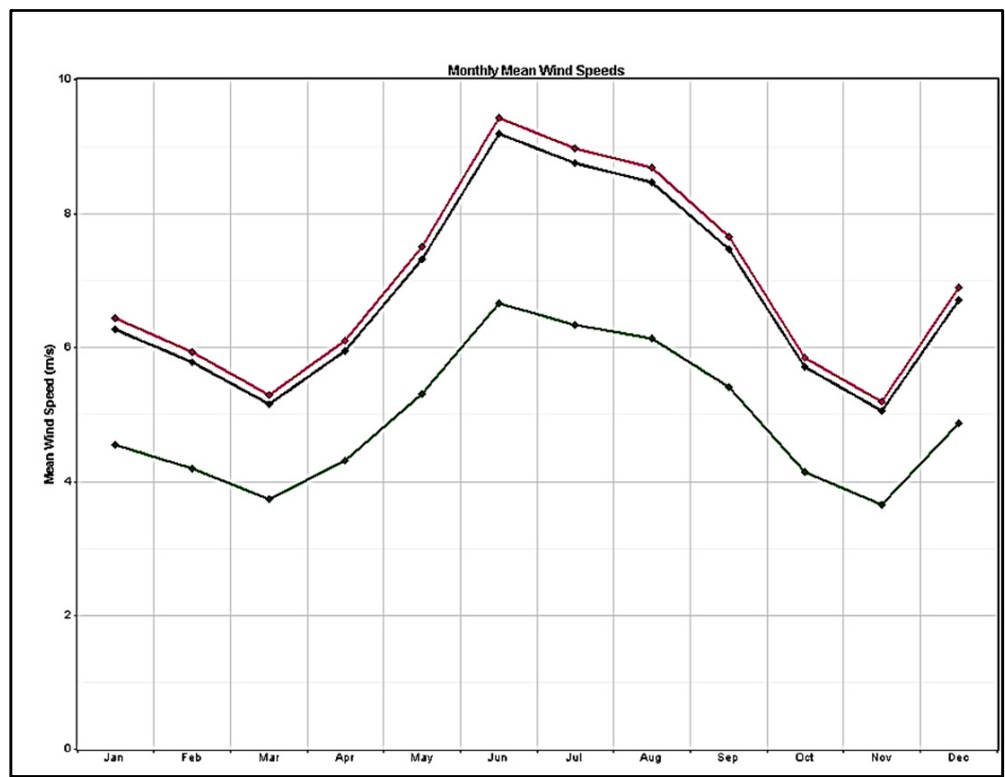

**Figure 21.** Monthly mean wind speed for the *Villa Teresa* station. Pink: 120 m; dark: 100 m; green: 10 m. Own elaboration.

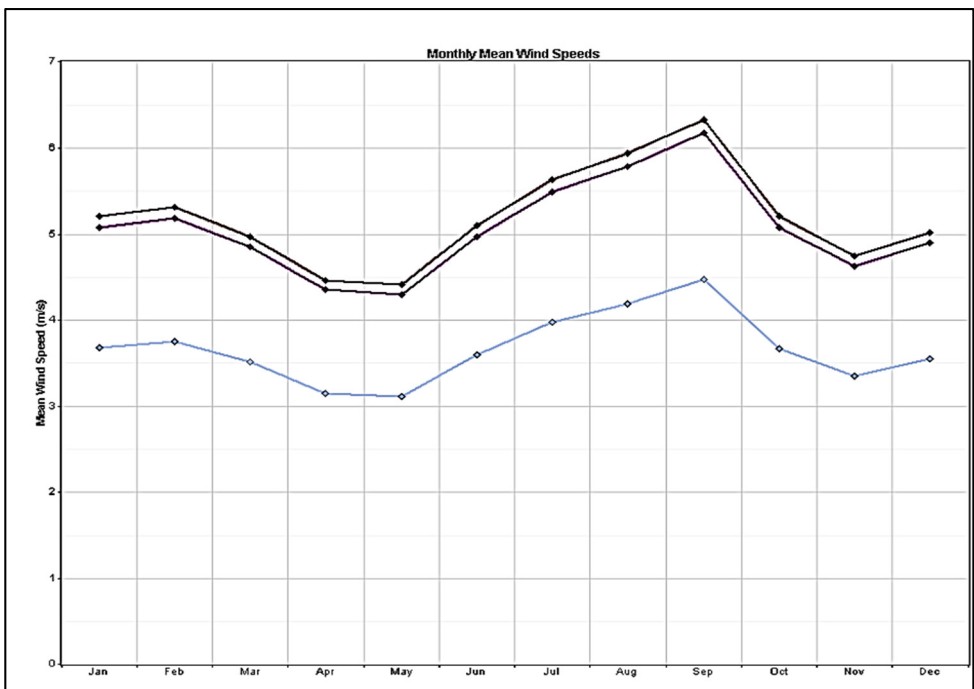

**Figure 22.** Monthly mean wind speed for the *Viento Libre* station. Brown: 120 m; purple: 100 m; blue: 10 m. Own elaboration.

*3.2. Site Density Air Determination*

For this calculation, the average temperatures of the areas of interest were consulted and the total height value of each location was determined, analyzing the contour lines of

our polygons (average altitude) and the height of the hub at which we will determine the wind fields: 120 m high.

In this procedure, Equation (1) was applied, and the contours of the emplacement were consulted in synchrony with the highest position in which the wind turbines are expected to be placed. Google Earth and QGIS tools were used for visual assessment. Although the intention was to include a zone of low altitude above sea level (higher temperature) such that the analysis would be much broader, the lack of stations, data, and sectors together with the stipulated selection conditions did not allow this.

The data used and the density values obtained through the *Wasp Air Density Calculator* tool are presented in Table 11.

**Table 11.** Calculated values for the air density at the indicated sites.

| Department | Station Name | Medium Temperature (°C) | Medium Altitude (m) | Total Height (m) | Site Density (kg/m³) |
|---|---|---|---|---|---|
| Boyacá | *Toquilla* | 9.6 | 2.800 | 2.920 | 0.887 |
| Cundinamarca | *Santa Cruz de Siecha* | 12.9 | 3.120 | 3.240 | 0.850 |
| Cauca | *El Tablazo* | 15.7 | 1.600 | 1.720 | 1.001 |
| Nariño | *Viento Libre* | 17.8 | 1.250 | 1.370 | 1.035 |
| Bogotá | *Villa Teresa* | 13.4 | 3.555 | 3.675 | 0.809 |

*3.3. Energy Simulations and Loss Scenario*

Using the Wasp tools Wind Climate Analyst and Turbine Editor, the .tab and .wtg files were obtained, which were loaded together with the .txt files of the location coordinates of each wind turbine (Google Earth) and the data from our series extrapolated to a height of 120 m (*Windographer*). The consolidated energy results of each proposed wind farm are presented below in Table 12.

**Table 12.** Consolidated energy production values.

| Station Name | Total Production (GWh) | Net Production (GWh) | Wake Loss (%) | Capacity Factor (%) | Maximum RIX (%) |
|---|---|---|---|---|---|
| *Toquilla* | 16.05 | 15.73 | 2 | 17.7 | 0 |
| *Santa Cruz de Siecha* | 10.33 | 9.93 | 3.79 | 8.5 | 0.2 |
| *El Tablazo* | 2.15 | 2.09 | 3.16 | 1.4 | 2.7 |
| *Viento Libre* | 93.53 | 91.96 | 1.68 | 28.1 | 13.3 |
| *Villa Teresa* | 131.48 | 129.79 | 1.29 | 29.5 | 7 |

It is important to mention that several test sites were assessed in each polygon to ensure that the wind turbines did not exceed the individual limit of losses above 8% (wake loss). This determination was applied in favor of having a future improvement and optimization ranges associated with the position and number of wind turbines, and with the coverage of the polygons of interest and the areas that can be included for other wind analyses in the country. The results from applying the loss scenario and Equations (4)–(7) are shown in Table 13.

**Table 13.** Normalized values with the loss scenario.

| Station Name | Net Production (GWh) | Time at Rated Power (h) | Capacity Factor (%) |
|---|---|---|---|
| *Toquilla* | 13.76 | 1.329 | 15.17 |
| *Santa Cruz de Siecha* | 8.69 | 629 | 7.18 |
| *El Tablazo* | 1.82 | 105.5 | 1.20 |
| *Viento Libre* | 80.47 | 1120.42 | 24.20 |
| *Villa Teresa* | 113.56 | 2992.35 | 34.16 |

Figures 23–27 represent the geographical layout of the proposed wind farms, with their respective contour lines in the established limits of each study polygon. These configurations ensure compliance with the good practices of the wind industry depending on the space, area, and parameters contemplated in this multifactor analysis.

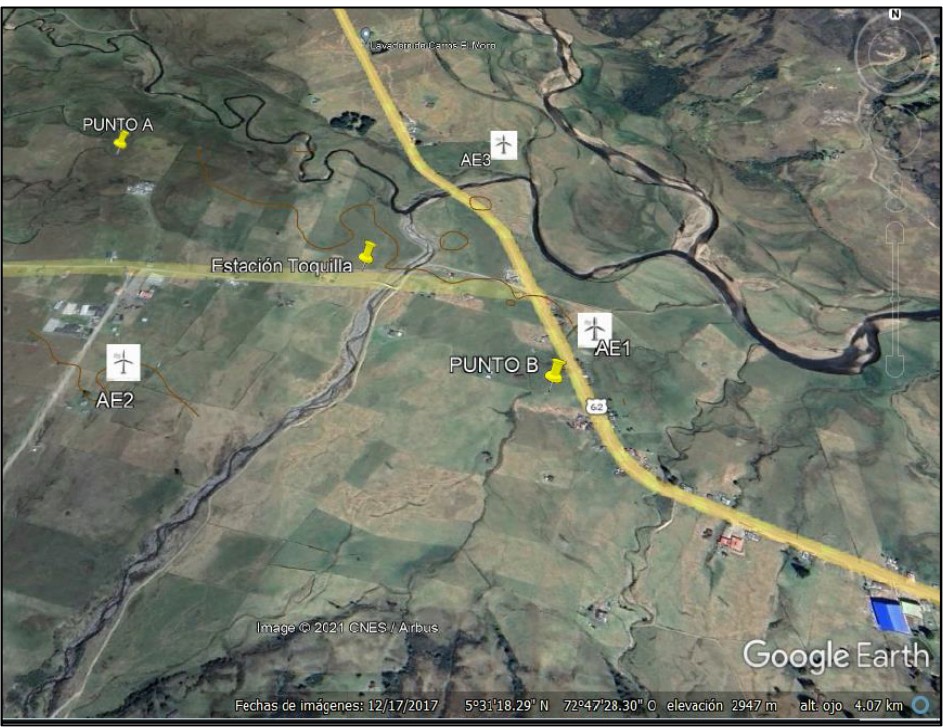

**Figure 23.** *Toquila* wind farm arrangement. Own elaboration.

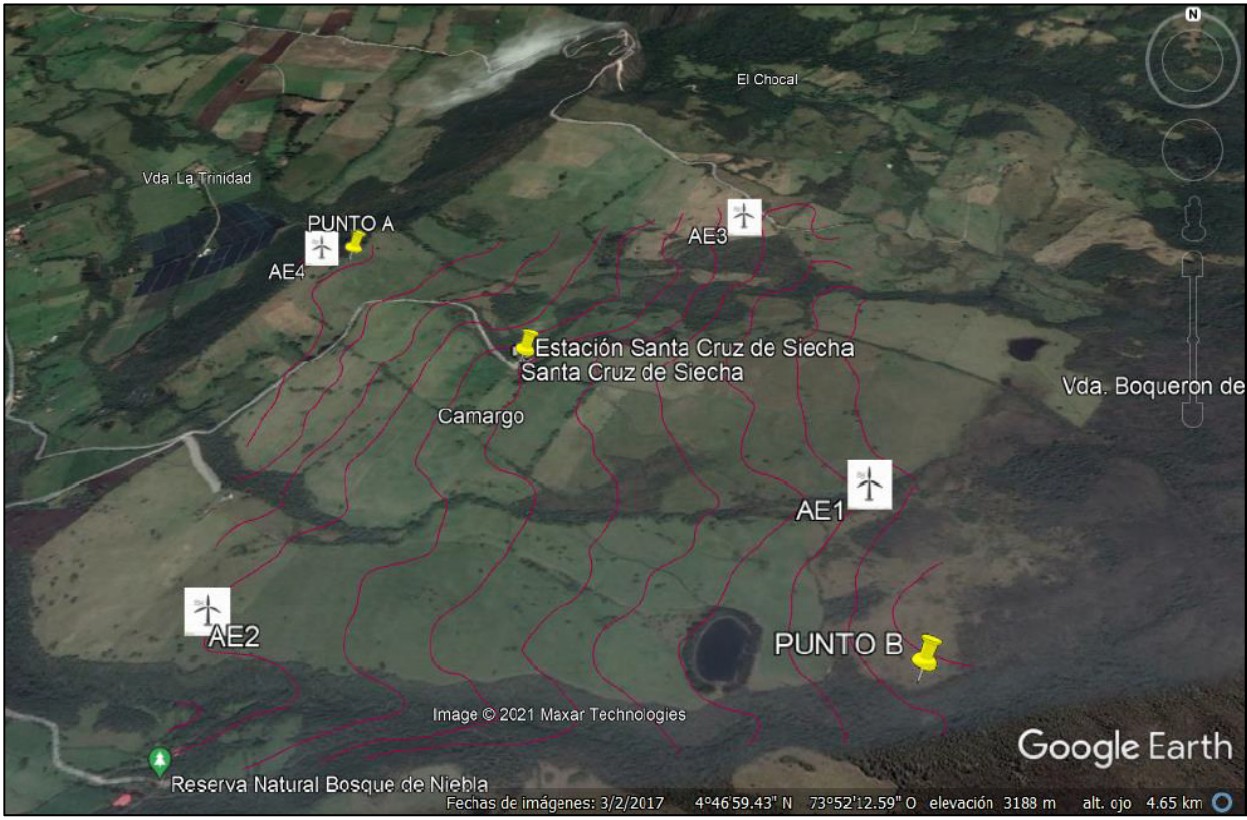

**Figure 24.** *Santa Cruz de Siecha* wind farm arrangement. Own elaboration.

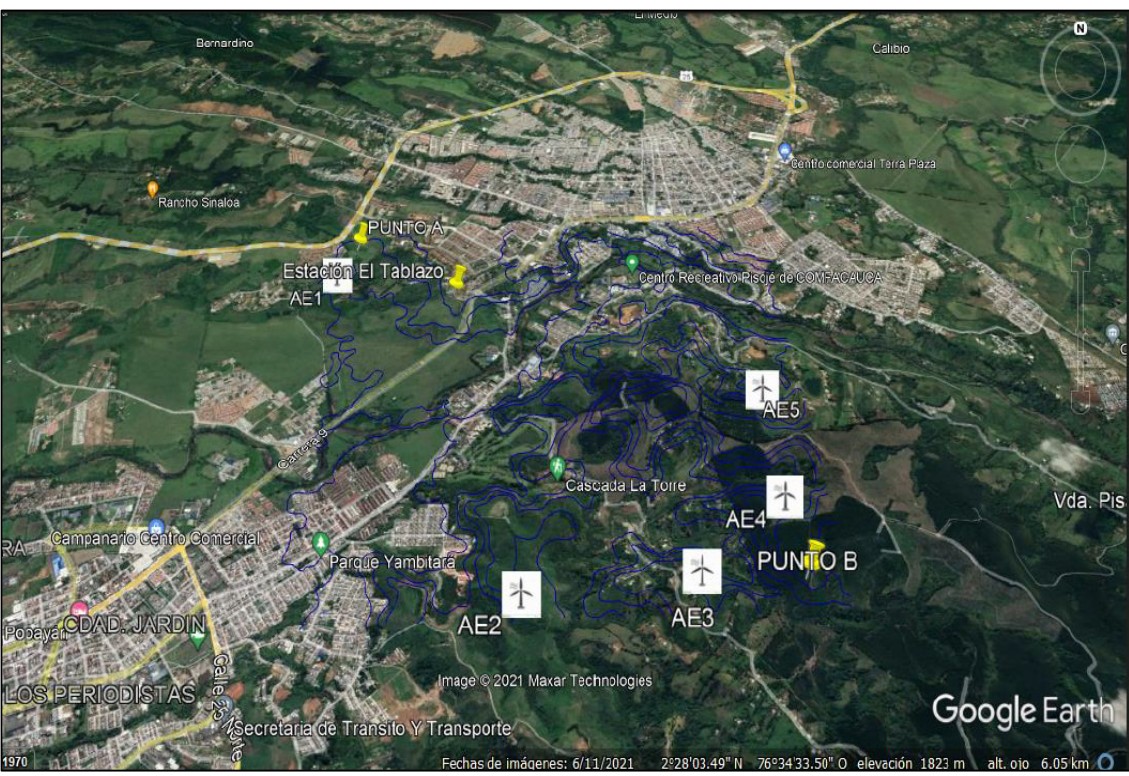

**Figure 25.** *El Tablazo* wind farm arrangement. Own elaboration.

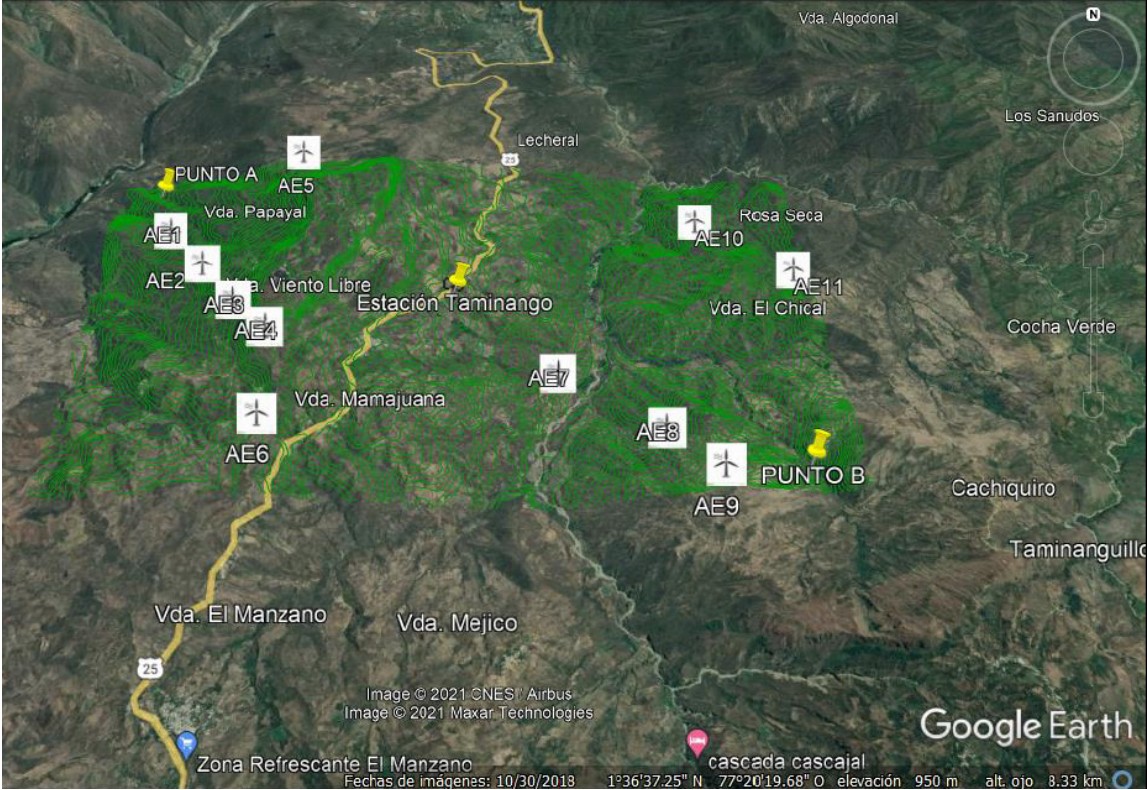

**Figure 26.** *Viento Libre* wind farm arrangement. Own elaboration.

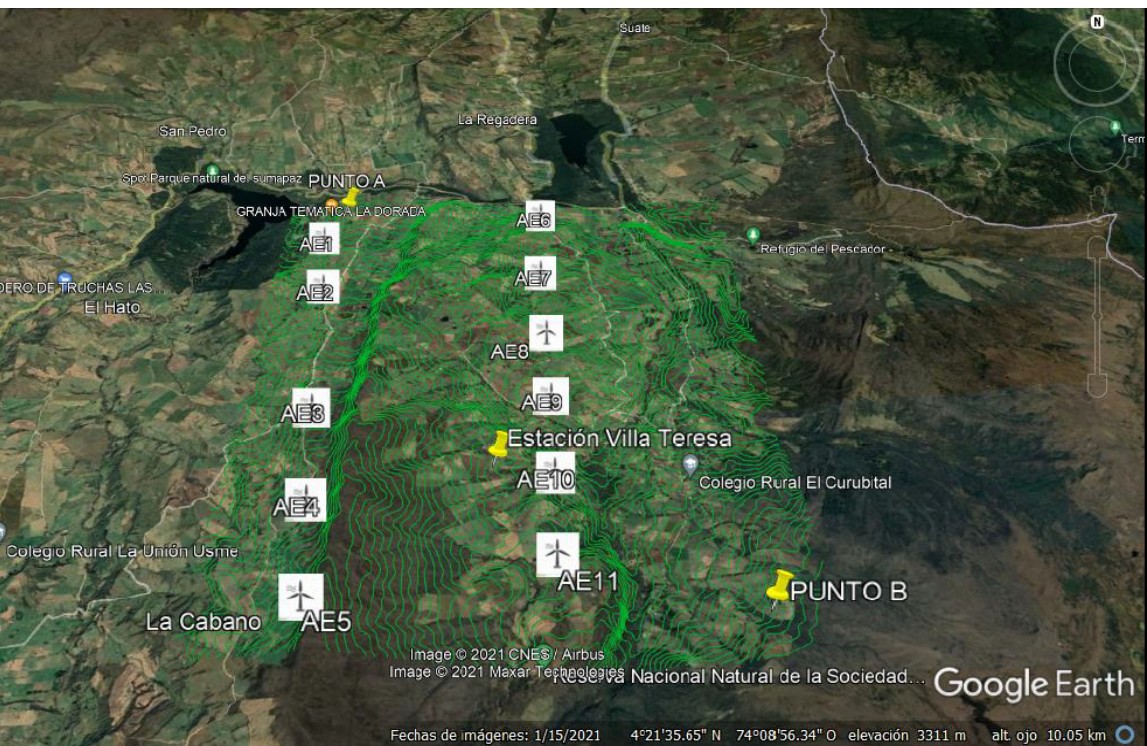

**Figure 27.** *Villa Teresa* wind farm arrangement. Own elaboration.

*3.4. Environmental and Economic Analysis*

In this section, the effects on four elements are reviewed: flora and fauna, soil and water resources, landscape, and urban populations and infrastructure.

Regarding the flora and fauna in Colombia, there is a high number of endemic species of flora and fauna, being a country with high standards in terms of biodiversity; therefore, it is not a minor detail to consider in the analysis of wind sites, since the greenery and ecosystems can be significantly altered.

From this analysis, it is possible to identify that the most vulnerable areas in this regard correspond to *Toquilla* and *Villa Teresa*, since they are habitat areas for endemic animal and plant species and that are in coexistence with bodies or streams of water with valuable characteristics for the sector. There is also a high impact in *Viento Libre* and *Santa Cruz de Siecha* due to the same aspects, but to a lesser extent.

The bodies of water and their springs are a very relevant item, since a possible effect can be perceived in the areas of interest, especially in *Villa Teresa*, since the frailejones and moors are vital for the maintenance of water reserves in the country, with the other areas not necessarily being exempt from a relevant effect.

For the landscape, in wind farms, it is a very difficult aspect to mitigate, since there will always be an effect, even if it is slight, in terms of noise, greenery, and pollution. However, the most affected area is *El Tablazo*, since the green areas that cross the urban corridor are affected.

In this regard, although all areas are affected, those with the least alteration may be *Viento Libre* and *El Tablazo*, since they are far from virgin areas of soil and tributaries, although all areas have a high impact associated with the loss of green landscapes. In the same way as in the landscape aspect, the *El Tablazo* area suffers substantially as it is less than two kilometers from the city, although the *Toquilla* area is also greatly affected. In the other areas, the effect is medium and high.

These concepts were obtained thanks to the satellite panorama of Street View in Google Earth, certain environmental studies specific to each area of interest, such as Municipal

Risk Management Plans (PMGR), Territorial Planning Plans (POT), and Planning and Management Plans of the Basin (POMCA), and government entities.

A rating matrix was generated for each of the four elements impacted at each of the sites, as reported in Table 14.

**Table 14.** Environmental impact risk matrix by wind farm.

| Station Name | Flora and Fauna | Soil and Water | Landscape | Urban Population and Infrastructure |
| --- | --- | --- | --- | --- |
| *Toquilla* | Very high | Very high | High | High |
| *Santa Cruz de Siecha* | High | High | High | Medium |
| *El Tablazo* | High | High | Very High | Very High |
| *Viento Libre* | High | Medium | High | Low |
| *Villa Teresa* | Very high | Very high | Very High | Medium |

Finally, regarding the economic aspect, Equations (8)–(10) were used. We started from an equal sale price for all cases of COP 420/kWh, about EUR 0.093/kWh (annual increase of 6%), since there are no diversified electricity market rates in Colombia, as is usually the case in the European Union, and an upward behavior was identified in the energy costs of the country for the year 2021.

The productive life scenario is 20 years at a discount rate of 2.5%, while the initial investment (CAPEX) and operating expenses (OPEX) were determined as follows for each site. All costs were determined in euros based on standard economic information of wind projects that have already been executed, but due to the geographical location, the costs of supplying wind turbines are overestimated.

For the project to be interesting, a positive internal rate of return of more than 3% is proposed. The following are the CAPEX and OPEX costs used, and the results are presented as a comparison between sites in Figures 28–30.

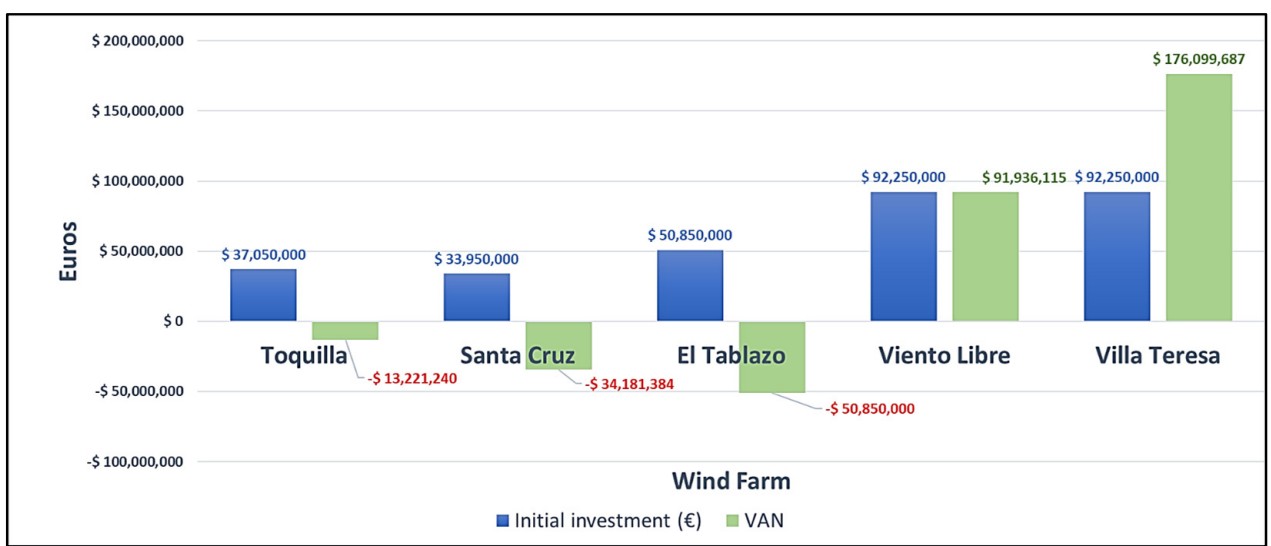

**Figure 28.** Initial investment with respect to VAN. Own elaboration.

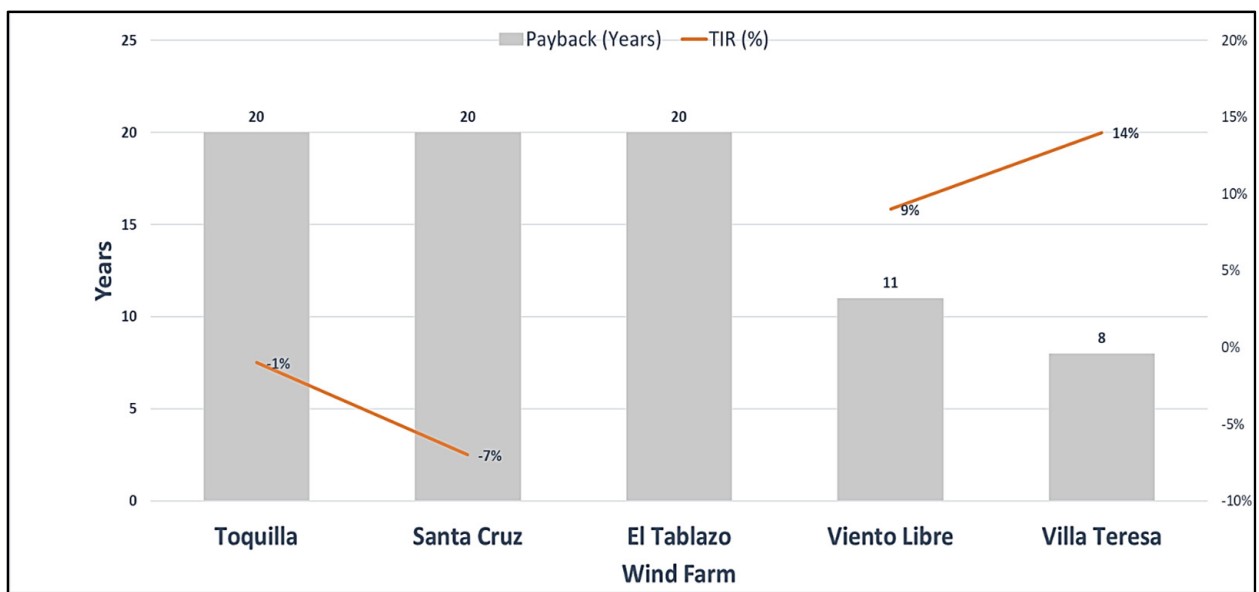

**Figure 29.** Payback and TIR for sites. Own elaboration.

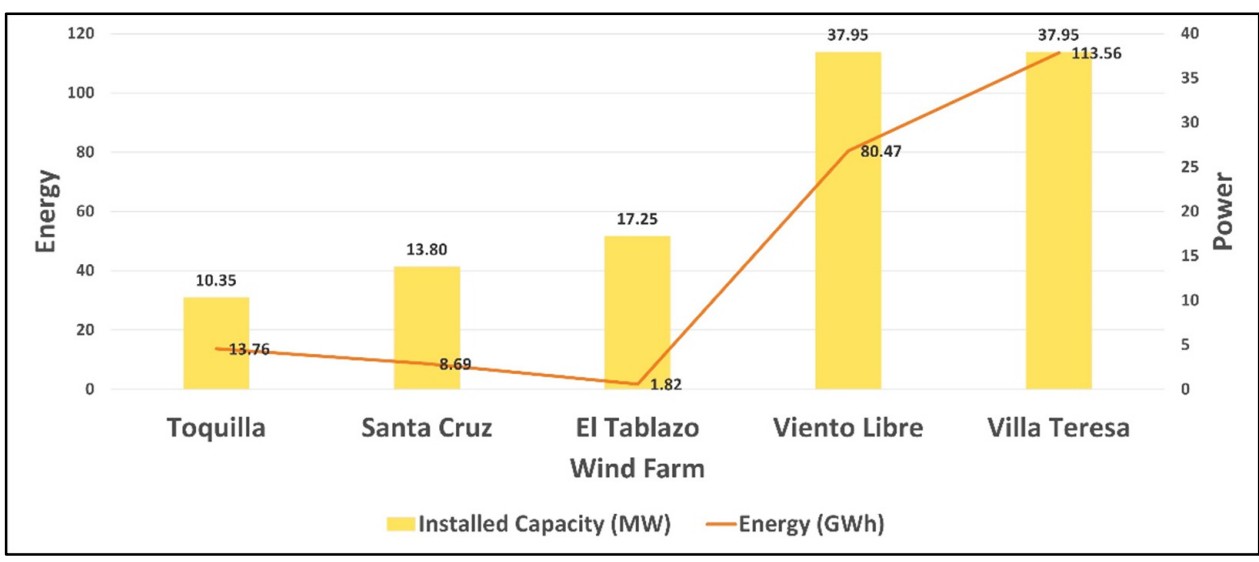

**Figure 30.** Produced energy with respect to the installed capacity. Own elaboration.

- Supply, transport, and assembly of wind turbines: EUR 2,000,000/MW;
- Wind farm civil works: EUR 5,500,000;
- Control tower civil works: EUR 1,000,000;
- Electrical supplies: EUR 8,200,000;
- Electrical installation: EUR 650,000;
- Development expenses in construction: EUR 1,000,000;
- Park operation and maintenance expenses: EUR 150,000/year (1% annual increase);
- Wind turbine operation and maintenance expenses: EUR 18,000/year (1% annual increase);
- Annual staff costs: EUR 45,000/year;
- Annual insurance and tax expenses: EUR 250,000/year (increments of EUR 1500 per year);
- Annual rent of land: EUR 2000/MW;
- Other expenses: EUR 15,000 per year (1% annual increase).

*3.5. Selection Matrix*

To provide an opinion regarding which project has the most benefits, a selection matrix was built from 14 associated factors (independent and compound). In accordance with what was mentioned in the Methods section, a rating model based on a simple Likert scale was built for various elements directly or indirectly linked to the multifactorial analysis.

These elements were assigned a numerical score of values between 1 and 5, with 1 being the lowest score and 5 the highest score, noting that for environmental and climatic factors, a higher score corresponds to a less affected site.

The results are shown in Table 15, in which *Viento Libre* is found to be a superior option based on the multifactorial analysis.

**Table 15.** Selection matrix results.

| Factor | *Toquilla* | **Santa Cruz** | *El Tablazo* | *Viento Libre* | *Villa Teresa* |
|---|---|---|---|---|---|
| Availability and Representativeness of Data | 4 | 3 | 5 | 5 | 3 |
| Measurement Campaign Feasibility | 2 | 4 | 1 | 5 | 3 |
| Site Altitude | 3 | 4 | 2 | 1 | 5 |
| Location | 3 | 4 | 1 | 5 | 2 |
| Possibility of Inclement Weather | 4 | 1 | 3 | 5 | 2 |
| Energy Discharged to Network | 3 | 2 | 1 | 4 | 5 |
| Economic Profitability | 3 | 2 | 1 | 4 | 5 |
| Environmental Impact | 1 | 2 | 1 | 3 | 1 |
| Access to National Interconnected System (SIN) | 3 | 1 | 5 | 4 | 2 |
| Wind Speed | 3 | 2 | 1 | 4 | 5 |
| Capacity Factor | 3 | 2 | 1 | 4 | 5 |
| % RIX Maximum | 5 | 4 | 3 | 1 | 2 |
| Power Density | 4 | 2 | 1 | 3 | 5 |
| Site Loss Percentage | 3 | 1 | 2 | 4 | 5 |
| **Final Score** | **44** | **34** | **28** | **52** | **50** |

## 4. Discussion

Given the results, it is necessary to compare the values obtained here to those of similar studies that have been carried out, especially where a series of factors and elements that tend to be multifactorial are weighted. In our case, based on Table 3 concerning related works, it was possible to compare certain elements that, although they differ in methods, allow a quantitative comparison. This is related to the reality indicated throughout the work, where it is emphasized that there are no studies of multifactorial characteristics in Colombia with similar methodologies that can be the subject of comparisons.

Table 16 compares the values of certain variables calculated in related research in comparison to our methodology, based on Table 3. Emphasis is placed on the results related to the Colombian territory, even without similar methodologies, and the reasons why there are numerical variations are discussed.

**Table 16.** Quantitative comparison of variables between the multifactorial analysis and similar research.

| Research | Favorable Locations (Department) | Capacity Factor (%) | Time at Rated Power (% of Year) | Installed Capacity (MW) | Energy (Gwh) |
|---|---|---|---|---|---|
| [11] | Guajira, Atlántico, San Andres | 90 | 8–70 | - | - |
| [12] | Boyacá | - | 21–55 | 20 | 15.6–88 |
| Multifactorial Analysis | Boyacá, Nariño, Cundinamarca, Cauca, and Sumapaz | 15–34 | 7–34 | 10–38 | 1.8–113 |

It is evident that the way of defining the locations used in the multifactorial analysis reduces the magnitude of the results, which is clearly linked to the disposition of the study polygons. Despite developing our own methodology, the possible mathematical determination of turbines, the application of more complex probabilistic methods, the possible contribution of a load analysis, and the application of two software packages generate a lack of variability in the results which can be seen reflected in the very low values of *El Tablazo* and the hopeful values of *Viento Libre* or *Villa Teresa*.

Although there are studies with similar characteristics worldwide, it cannot be ignored that the Colombian reality is far below countries such as Spain or Germany; therefore, a comparison of energy and site conditions would not be equitable. The foregoing is maximized by having eliminated the area of La Guajira in this work, the most prospective and analyzed in the country, meaning that any higher energy value could only be found in that area or in the offshore areas of influence (also excluded). This, however, was carried out in favor of characterizing the internal areas of the country, which have been little reviewed to date.

On the other hand, the sizing of wind farms mostly uses multicriteria decision making (MCDM) methods. However, due to the number of factors selected, the conditions of exclusion of areas, and the unique procedure that was used in the handling of the data and their analysis, a Likert scale was used, but this does not imply that the power of the decision and the refinement of the results provided by MCDM methods were not dimensioned.

As there are so many sub-areas of mathematical application linked to wind energy, it is very common not to address all the topics in the same work, which is why it is contemplated to execute a validation of the site selection matrix using the AHP and TOPSIS methods.

## 5. Conclusions

Colombia has alternative wind potential, different from that provided by the Alta Guajira area. The wind opportunities identified in this analysis correspond to areas where sites can be built that contribute to the energy matrix, diversifying it and leaving aside the dependence on hydraulic dams. This is supported by the promising energy results of *Viento Libre* and *Villa Teresa*, without discarding the areas with lower ratings after applying the selection matrix, since they are susceptible to regional polygonal optimizations.

Measurements at a height of 10 m are clearly impacted by the obstacles close to each measurement station, so it is most advisable to propose a tailored measurement campaign in these analyzed areas, since doing so may result in more accurate direction and velocity data. Therefore, together with load analysis and an attractive economic project, the wind farm options in *Viento Libre* and *Villa Teresa* could be implemented, but with the associated environmental limitations mitigated in the latter, which must be considered as exclusionary.

The results reaffirm that it is not always possible to establish a direct relationship between the projected highest income to be generated and the most appropriate technical implementation option, since there are some factors that can cause a change in prevalence. It is also clear that even a few correctly located wind turbines can generate better results than several in a less than optimal location, as can be seen in the generation of 13.76 GWh at *Toquilla* (three wind turbines) and 1.82 GWH at *El Tablazo* (five wind turbines), with both even being totally discounted economically due to their null recovery of investment.

Despite constructing a selection matrix with numerous (14) elements, all the possible additional factors that may be relevant such as roughness or the standard deviation of the data for turbulence effects were not covered in *Windographer*, which leaves open the possibility of optimizing the results. At this point, as mentioned in the Discussion section, it is evident that it is necessary to apply MCDM methods and compare the selection obtained by the original matrix with respect to a numerical model.

It is recommended that this same analysis be applied to the zones of high offshore potential identified in the IDEAM wind map, since, as evidenced in the Wind Atlas, the Colombian Pacific and Atlantic regions are highly favorable, and it is a waste not to consider more wind energy projects and research in the country. It is not new that all the projects under development and those that have not yet been awarded in Colombia have La Guajira as their focal point, but works such as the one presented here show that it is feasible to consider other regions including starting validations in offshore locations.

**Author Contributions:** Conceptualization, I.C.G.-G. and A.R.-C.; methodology, I.C.G.-G.; formal analysis, A.R.-C.; resources, I.C.G.-G. and A.R.-C.; data curation, A.R.-C.; writing—original draft presentation, A.R.-C.; writing—review and editing, I.C.G.-G. and A.R.-C.; supervision, I.C.G.-G. All authors have read and agreed to the published version of the manuscript.

**Funding:** This research received no external funding.

**Institutional Review Board Statement:** Not applicable.

**Informed Consent Statement:** Not applicable.

**Data Availability Statement:** The data presented in this study are available on request from the corresponding author.

**Conflicts of Interest:** The authors declare no conflict of interest.

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
