# Peer review of "Multifactorial Analysis to Determine the Applicability of Wind Power Technologies in Favorable Areas of the Colombian Territory"

_2674-032X, doi:10.3390/wind2020020_

Round 1

Reviewer 1 Report

Following are the comments:

  1. Authors should enrich the Introduction section by discussing following articles
    1. H. Dhiman and D. Deb, "Wake Management based Life Enhancement of Battery Energy Storage System for Hybrid Wind Farms," Renewable & Sustainable Energy Reviews, Elsevier, 2020 (SCI, Impact Factor: 10.556). Accepted

    2. H. Dhiman, D. Deb, A. M. Foley, "Lidar assisted wake redirection in wind farms: A data-driven approach," Renewable Energy (Elsevier), Volume 152, June 2020, Pages 484-493 (SCI, Impact Factor: 5.439). https://doi.org/10.1016/j.renene.2020.01.027

    3. H. Dhiman, D. Deb, J. Guerrero, "Hybrid Machine Intelligent SVR Variants for Wind Forecasting and Ramp Events, "Renewable and Sustainable Energy Reviews, Elsevier, Volume 108, July 2019, Pages 369-379, (SCI, Impact Factor: 10.556). https://doi.org/10.1016/j.rser.2019.04.002

    4. H. Dhiman, D. Deb, V. Muresan, M. L. Unguresan, “Multi-Criteria Decision Making Approach for Hybrid Operation of Wind Farms, Symmetry MDPI, 11, 675 (SCI, Impact Factor: 2.143), 2019. doi:10.3390/sym11050675

    5. H. Dhiman, D. Deb, V. Muresan, V. E. Balas, "Wake Management in Wind Farms: An Adaptive Control Approach, " Energies, MDPI, 12(7), 1247 (SCI, Impact Factor: 2.707), 2019. https://doi.org/10.3390/en12071247

  2. Discuss the possibility of using MCDM techniques for determining best suited turbine generators in wind farms of Columbia
  3. Authors should plot wind speed time-series charts along with histogram depicting the distribution of wind speed across different wind farms
  4. Apply any ONE (TOPSIS, COPRAS, SAW or AHP) as a MCDM method for site selection

Author Response

Dear reviewer, we appreciate your work in reviewing our article. We attach a file with our response. Best regards,

Reviewer 2 Report

The authors write the article very well.  I have no comments. 

Author Response

The authors express their sincere appreciation to the Reviewer #2.

Reviewer 3 Report

None of the Equations have citation. This problem must be resolved.

Novelty and research gap of the work must be clearly explained.

English must be refined too.

Otherwise, the paper seems to be good for this journal to be published.

Author Response

(The authors gave the same response as above.)

Reviewer 4 Report

At present, the global wind power capacity continues to increase, but the development of wind power in Colombia is stagnant. The authors propose a viable solution for wind power plant options, by analyzing the climatic, geographical, Socio-environmental and economic factors of different areas of Colombia, and give the selection of favourable zones and their sensitivities, this paper is of great significance to the development of wind power in Colombia.

The paper is well-written and straightforward. Methods are well explained and fit with the objectives proposed. Discussion and conclusions need to be more in-depth. The language may better be polished up.

I recommend that it should be accepted with some revisions.

In general, my primary comment on the manuscripts is:

1. Lines 129, 186…, the numbers are wrong.

  1. Lines 204, Figure 3. The presentation of this picture is not direct enough to illustrate the relationship between wind speed and power generation Feasibility.

It is better to put the words in the picture upright and clearly correspond to different wind speed ranges.

  1. lines 708-709 “as evidenced in the Wind Atlas, the Colombian Pacific and Atlantic regions are too attractive not to be taken advantage of in the future.” The expression seems ambiguous.

The discussion in the entire conclusion paragraph is not deep enough, and it is suggested to add a more specific summary and specific recommendations for the future development of wind power in Colombia.

  1. Chapters 2 and 3 provide a detailed analysis of the characteristics of wind energy across different factors and regions. Table 14 summarizes the selection matrix results. However, only a 1-5 scale is used for the final evaluation. The relationship between the analyses and summary is very weak and not mathematical. It is recommended that in each factor discussion of Chapter 3, the influence of factor can be quantified for each region, and finally summarized in Table 14, instead of simply choosing from 1-5 points.

Author Response

(The authors gave the same response as above.)

Reviewer 5 Report

The article "Multifactorial Analysis to Determine the Applicability of Wind Power Technologies in Favorable Areas of the Colombian Territory" has an interesting topic and to publish some points should be observed:
1. The research literature section should be added. Also, the conducted studies are summarized in a table.
2. The quality of the Figures is low and needs to be improved.
3. The Discussion section should be added to the article and in this section, the results should be compared with previous studies and the results should be analyzed.
4. Please, examine the article for English language structure.

Author Response

(The authors gave the same response as above.)

Round 2

Reviewer 1 Report

Authors have carried out revision adequetly. 

Reviewer 4 Report

I'm pleased that the authors provide a more complete discussion in this paper, which provides strong guidance for the development of the wind industry.

Given all the comments of the reviewers, the author makes detailed modifications. I think the author's revision has met all the main opinions of reviewers.

Overall, the Arduous and meaningful work deserves to be published in this round.

Reviewer 5 Report

The authors answered most of the concerns.